# Webis: A Multi-Level Pipeline Integrating Node Classification, Reverse-Coloring, and Semantic Pruning for Web Content Extraction

## Abstract

Web content serves as a primary data source for large-scale pre-trained language models and web applications such as search engines and recommendation systems, making its quality crucial. However, modern web pages often contain substantial structural and semantic noise, leading to unstable model training and semantic deviations that impair downstream performance. Existing methods show significant limitations in handling complex web noise, frequently failing to process interactive elements, dynamic advertisements, and multimodal content. To address these challenges, we propose Webis, a multi-level framework for web content extraction that integrates node classification, subtree reverse-coloring, and semantic pruning to refine both local and global semantics. Experiments show that Webis improves both node-level noise detection and end-to-end extraction quality. Extensive experiments verify that Webis combines the precision of neural models with the scalability of engineering tools. On SWDE-small, it achieves an 82.2 node-level F1 score, surpassing GPT-4o by a significant margin of +37.0. On the large-scale SWDE-huge benchmark, Webis attains a 99.6% execution success rate, matching the execution robustness of mature text extraction libraries such as Trafilatura, while delivering superior extraction quality (+1.9 CEQI over the strongest heuristic baseline).

## 1. Introduction

With the rapid expansion of the Internet, web pages remain a primary channel for information acquisition (Al-Ghuribi & Alshomrani, 2013; Whang et al., 2023). However, modern web pages embed substantial boilerplate and dynamic elements, which reduce content density and hinder both user reading and downstream applications such as crawlers (Barbaresi, 2016; Uma & Latha, 2019). Meanwhile, LLMs increasingly rely on large-scale, high-quality web corpora, making low-noise content extraction a critical prerequisite for dataset construction (Peters & Lecocq, 2013; Cai et al., 2022; Baunsgaard & Boehm, 2025).

Web-based text is a key data source for LLM training, yet robust extraction at scale remains difficult under cost and efficiency constraints. Existing tools mainly rely on heuristic cues (HTML structure, text density) or traditional ML, and often break down on complex or sparse pages—leading to missing information, residual noise, and fragmented text.

To address multi-level noise, we propose **Webis**, a three-layer extraction framework combining node classification, subtree reverse-coloring, and semantic pruning. After removing non-content elements (e.g., CSS/scripts/comments) and collapsing redundant tags, Webis predicts noisy DOM nodes using a lightweight classifier with structural–semantic features. A bottom-up reverse-coloring "merge–prune" procedure then refines subtrees to improve global coherence, and the retained nodes are reassembled to preserve the original reading flow. Finally, a large pre-trained language model performs paragraph-level filtering to remove residual noise.

Webis improves both purity and integrity of extracted text. We further introduce CEQI to quantify extraction quality. Experiments show that Webis achieves **82.4%** node-level accuracy, surpassing ChatGPT-4 and DeepSeek by **28.9%** and **33.4%**, with recall gains of **29.7%** and **32.7%**, respectively. Compared with common web extraction methods used in current LLM pipelines, Webis improves average precision by **6.0%**, recall by **16.5%**, and CEQI by **11.5%**, demonstrating better accuracy, noise reduction, and semantic integrity for large-scale corpus construction.

The main contributions are:

- **Multi-level noise localization with structural consistency.** A lightweight classifier fusing node features and a bottom-up reverse-coloring strategy for node/subtree-level denoising.

- **Semantic-level filtering.** Paragraph-level re-evaluation

[1]Anonymous Institution, Anonymous City, Anonymous Region, Anonymous Country. Correspondence to: Anonymous Author <anon.email@domain.com>.

Preliminary work. Under review by the International Conference on Machine Learning (ICML). Do not distribute.

using LLMs to remove residual promotional/advertising noise and flag low-confidence decisions.

- **System, evaluation, and open-source.** A three-stage implementation validated on a ClueWeb22-derived DOM dataset and SWDE (SWDE-small/SWDE-huge), outperforming multiple baselines; code and documentation released.

## 2. Background and Related Work

Web pages often intermix valuable text with boilerplate elements such as navigation bars, advertisements, and copyright notices, which complicates automated extraction and corpus construction. Early approaches used handcrafted, rule-based pattern matching to separate content from noise by leveraging HTML tags and layout cues (Wang et al., 2009; Yi et al., 2003). Although effective on narrowly scoped sites (Das et al., 2012; Li et al., 2024), these methods are brittle: small layout changes can invalidate rules and lead to high maintenance costs (Kohlschütter et al., 2010; Whang et al., 2023).Notable approaches also explored unifying tree structures into forests to handle diverse data records (Hao et al., 2011)

To improve adaptability, researchers moved to DOM-tree analysis and filtered noise using node-level structural features such as depth, child count, and text length (Lin & Ho, 2002; Yi et al., 2003; Kohlschütter et al., 2010). While more flexible than static rules (Sun et al., 2011; Gupta et al., 2003), structure-only methods still generalize poorly across diverse modern layouts. Later work incorporated machine learning by combining structural descriptors with textual signals (Chau & Chen, 2008; Zhou & Mashuq, 2014), enabling classifiers to separate main content from boilerplate with less manual effort and stronger cross-site generalization (Peters & Lecocq, 2013; Mehta & Narvekar, 2015).

The advent of deep learning further enriched this landscape (Htwe & Hla, 2010). Convolutional and recurrent neural networks enable end-to-end extraction by learning representations directly from HTML or rendered page snapshots (Cai et al., 2003; Vogels et al., 2018). Vision-based approaches process page screenshots with CNNs to identify visually salient content blocks, achieving strong results in benchmark evaluations (Hiremath & Algur, 2009; Liu et al., 2009; Gogar et al., 2016). Practical tools such as Readability and Trafilatura also validate the efficacy of hybrid and neural-driven schemes.

More recently, large pre-trained language models have offered unprecedented semantic understanding and adaptability for content extraction tasks (Wang et al., 2022; Havrilla & Iyer, 2024; Penedo et al., 2024). Tools such as FireCrawl harness these models for context-aware filtering and reconstruction (Mendable.ai, 2019; Hofmann et al., 2025), yet they still face limitations such as high costs and a lack of structural constraints (Das et al., 2012).

Building on prior work, we present a low-noise extraction pipeline that combines DOM-aware node classification with structure-preserving subtree consolidation and lightweight semantic pruning. Compared with static rule-based engines such as Readability and Newspaper3k, our approach explicitly couples structural cues with semantic signals and performs multi-stage denoising at different granularities. This design targets higher content purity and coherence while ensuring high content purity and structural integrity.

## 3. Webis

### 3.1. System Overview

Webis is a multi-level content extraction pipeline for large-scale web data, aiming to achieve high-quality and robust text extraction in real-world environments with coexisting structural and semantic noise. It is a hierarchical framework realizing denoising and information integration from structure to semantics.

The overall workflow (Figure 1) is as follows: Raw webpages are first parsed into DOM trees, then sequentially processed by three core modules—Node-Level Extraction, Subtree-Level Refinement, and Semantic-Level Pruning. These modules form a bottom-up collaborative process for cross-level noise suppression and semantic fusion, ultimately generating structured, indexable, and training-ready textual content through a "from nodes to semantics" multi-dimensional purification.

- **Node-Level Extraction.** Webis conducts initial node-level cleaning via a lightweight classifier, which predicts DOM node labels by fusing structural features and semantic features for accurate, effective noise identification. The output is a structured subtree with high-confidence content nodes, laying the foundation for subsequent refinement.

- **Subtree-Level Refinement.** Building on node-level results, Webis applies a *Reverse-Coloring* strategy to aggregate locally coherent subtrees bottom-up, correcting fragmentation, over-pruning, and discontinuities from the previous stage. With structural connectivity and semantic consistency constraints, it reconstructs complete local paragraphs to enhance readability and contextual continuity.

- **Semantic-Level Pruning.** Finally, Webis performs global semantic pruning using LLM to evaluate the topical coherence of merged paragraphs and identify residual noise. This stage ensures the final output maintains global semantic coherence and textual purity, suitable

for downstream tasks like indexing, corpus construction, and web-scale knowledge extraction.

### 3.2. Node-Level Extraction: DOM Feature Engineering and Noise Prediction

To ensure robust handling of dynamic content, we first utilize a headless browser to fully render the webpage, executing JavaScript to materialize client-side generated elements. Subsequently, before extracting DOM features, we perform a near-lossless preliminary cleaning on this rendered DOM tree. This process removes non-semantic elements (e.g., CSS, scripts, comments, and meta tags) and prunes empty or redundant nodes, yielding a structurally consistent DOM tree for subsequent feature extraction.

During noise-node detection, we extract complementary structural and semantic features via *DOM-Validator*, and adopt a dual-view encoding scheme ⟨DOM-structure, textual-content⟩ to represent each node. Implementation details (dataset construction, DOM-Validator statistics, and classifier setup) are provided in Appendix D.

This work focuses on two core feature categories (derived from DOM-tree structured information):

1. **Structural Features**: XPath path, depth, HTML tag type, sibling/adjacency relations, and path-level priors (Risk-Tag and Noise-Confidence).

2. **Semantic Features**: Node text content and content–context cues encoded jointly with the XPath path.

#### 3.2.1. DEPTH-FIRST-SEARCH BASED NODE FEATURE EXTRACTION

In preprocessing, we traverse the DOM tree via depth-first search (DFS), skipping non-content subtrees (e.g., `<script>`/`<style>`/`<meta>`). For text-bearing nodes, we record the text payload, a normalized DOM path, and an ordinal index among same-tag siblings to ensure unique paths (supporting later subtree operations and text reconstruction). Formal definitions of the feature composition and the resulting node vector are given in Appendix B.

#### 3.2.2. RISK-TAG GENERATION

To provide a tag-level prior, we estimate a *Risk-Tag* probability $p_t$ for each HTML tag $t$ from corpus-level noise statistics. To address frequency imbalance, we use Bayesian smoothing for high-frequency tags and a global–local estimator for sparse tags.

**High-frequency tags.** For high-frequency tags with sufficient observations, we apply Bayesian smoothing with an

adaptive Beta prior. Let $N_t^{\text{total}}$ denote the total number of occurrences of tag $t$ in the corpus, and $N_t^{\text{noise}}$ denote the number of times nodes of tag $t$ are labeled as noise. We introduce a smoothing strength $s = \log_2(|\mathcal{T}|)$, where $\mathcal{T}$ is the set of distinct HTML tags observed in the corpus. The smoothed noise probability of tag $t$ is then computed as

$$p_t = \frac{N_t^{\text{noise}} + s}{N_t^{\text{total}} + 2s}. \tag{1}$$

**Low-frequency tags.** For low-frequency tags, the local estimate can be unstable, so we regularize it using a global prior computed over the entire tag set. The global noise prior is defined as

$$\pi_g = \frac{\sum_{t \in \mathcal{T}} N_t^{\text{noise}} + \alpha_g}{\sum_{t \in \mathcal{T}} N_t^{\text{total}} + \alpha_g + \beta_g}, \tag{2}$$

where $\sum_{t \in \mathcal{T}} N_t^{\text{noise}}$ and $\sum_{t \in \mathcal{T}} N_t^{\text{total}}$ are the total noise count and total occurrence count aggregated over all tags, respectively. The global smoothing parameters are set as $\alpha_g = 0.001 \sum_{t \in \mathcal{T}} N_t^{\text{total}}$ and $\beta_g = 2\alpha_g$.

Next, we compute a locally smoothed estimate using the same $s$,

$$\pi_{\text{local}} = \frac{N_t^{\text{noise}} + s}{N_t^{\text{total}} + 2s}, \tag{3}$$

and fuse the global and local estimates to obtain the final probability for sparse tags:

$$p_t = 0.3\,\pi_{\text{local}} + 0.7\,\pi_g. \tag{4}$$

Finally, the resulting $p_t$ values are discretized into three risk levels ([H]/[M]/[L]), and the corresponding label is attached to each DOM node as a tag-level prior.

#### 3.2.3. NOISE-CONFIDENCE CALCULATION

We further compute a node-level noise-confidence score $C$ by aggregating path risks into a baseline estimate, applying dynamic penalties that account for tag diversity and node depth, and normalizing the result to $[0, 1]$. Full mathematical derivations are provided in Appendix A.

After extracting feature vectors $\{\mathbf{x}_i\}_{i=1}^N$ for all DOM nodes and fine-tuning our denoising model, we perform node-level noise prediction. Given a training set $\{(\mathbf{x}_i, y_i)\}_{i=1}^N$ with binary labels $y_i \in \{0, 1\}$ (0 = clean, 1 = noise), we optimize

$$\mathcal{L}(\theta) = \sum_{i=1}^N \ell(y_i, \hat{y}_i) + \sum_{k=1}^K \Omega(f_k), \tag{5}$$

where $\ell(\cdot)$ measures the discrepancy between true labels and predictions, and $\Omega(\cdot)$ regularizes model complexity to

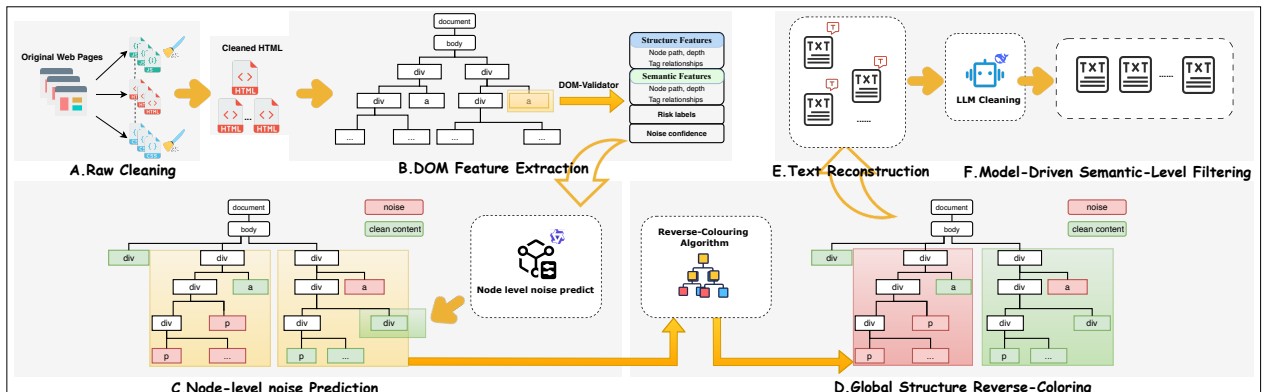

Figure 1. Overview of the proposed Webis framework for web content denoising.

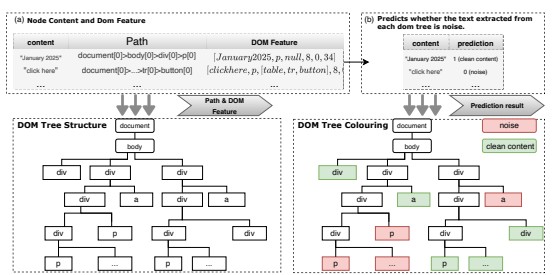

Figure 2. Illustration of node-level noise prediction and DOM-tree marking: extract the predicted node position and map the prediction results back onto the DOM tree.

prevent overfitting. At inference, each node $v_i$ receives a noise probability

$$p_i = P(y_i = 1 \mid \mathbf{x}_i). \tag{6}$$

### 3.3. Subtree-Level Refinement: Reverse-Coloring and Text Reconstruction

While node-level prediction is effective at identifying fine-grained noise patterns, isolated misclassifications can easily disrupt the structural coherence of extracted content due to the deeply nested and heterogeneous nature of web pages. To address this issue, we introduce a subtree-level refinement stage that enforces global consistency over local predictions by operating directly on the DOM hierarchy.

Let a webpage be represented as a DOM tree $T = (V, E)$, where each node $v \in V$ is associated with a node-level noise prediction $y_v \in \{0, 1\}$.

For a subtree $\tau_n$ rooted at node $n$, we define the noise ratio as:

$$R(\tau_n) = \frac{|\{v \in \tau_n : y_v = 1\}|}{|\tau_n|}. \tag{7}$$

Two thresholds are used to control subtree decisions: a merging threshold $T_1$ and a pruning threshold $T_2$. To mitigate

overemphasis from deeply nested subtrees, thresholds are adjusted adaptively with respect to depth $d$:

$$T_d = T - \alpha \cdot d, \tag{8}$$

where $\alpha$ controls the decay rate.

The reverse-coloring process consists of two complementary phases: a **Merge Phase** and a **Prune Phase**. In the merge phase, subtrees with a low overall noise tendency ($R(\tau_n) < T_1$) are treated as clean content blocks, even if they contain a small number of nodes predicted as noise, thereby correcting false-positive errors. In the prune phase, subtrees dominated by noisy content ($R(\tau_n) \geq T_2$) are removed entirely to eliminate residual false negatives.

After subtree-level refinement, text reconstruction is performed on the refined DOM tree to recover the original reading order and hierarchical layout, producing the final plain-text output. This step ensures that structural coherence and readability are preserved after noise removal. The detailed algorithmic procedures are provided in Algorithm 1 and Algorithm 2.

### 3.4. Semantic-Level Pruning: LLM-Driven Global Noise Filtering

After the completion of node-level screening and subtree-level reverse-coloring, the pipeline transitions from structural refinement to global semantic purification. At this stage, a large-scale pre-trained language model (ChatGPT-4o-mini) is employed to perform *context-aware, document-level noise filtering*, which extends beyond the capability of heuristic or feature-based models.

Noise removed in earlier stages is referred to as **Level-1 noise**, representing explicit structural impurities such as advertisements, scripts, and decorative widgets. However, some latent nodes—whose structures and textual forms closely resemble genuine body text—remain undetected. These nodes deviate semantically from the page's central

**Algorithm 1** Reverse Coloring Algorithm

**Require:** DOM tree $D_1$, merging threshold $T_1$, pruning threshold $T_2$, node-level predictions $y$
1: **Merge Phase:**
2: **for** each node $n$ from leaves to root **do**
3:    Compute noise ratio $R(\tau_n)$
4:    **if** $R(\tau_n) < T_1$ **then**
5:       Mark entire subtree $\tau_n$ as clean
6:    **end if**
7: **end for**
8: Obtain intermediate tree $D_2$
9: **Prune Phase:**
10: **for** each node $n$ in $D_2$ from leaves to root **do**
11:    Compute noise ratio $R(\tau_n)$
12:    **if** $R(\tau_n) \geq T_2$ **then**
13:       Prune subtree $\tau_n$
14:    **end if**
15: **end for**
16: **Return** refined DOM tree $D_3$

---

**Algorithm 2** DFS-Based Text Reconstruction

**Require:** Refined DOM tree $D_3$, original DFS order $S_{\text{dfs}}$
1: Initialize output buffer $B \leftarrow \emptyset$
2: **for** each node $n$ in $S_{\text{dfs}}$ **do**
3:    **if** $n \in D_3$ **then**
4:       Append cleaned text of $n$ to $B$
5:    **end if**
6: **end for**
7: **Return** concatenated plain text

---

theme and are thus defined as **Level-2 noise**. Because they often appear under standard HTML tags (e.g., <p>, <ul>) or within deeply nested hierarchies,they can bypass local classifiers. Identifying such content requires a model capable of holistic document-level semantic reasoning.

To this end, Webis incorporates an LLM-guided two-stage reasoning process:

- **Global Topic Comprehension.** The model first reads the entire page to capture its overarching theme and discourse structure, forming a global contextual representation that reflects domain intent—for example, the algorithmic logic of a technical document or the chronological narrative of a news article.

- **Paragraph-Level Semantic Filtering.** Each paragraph is then evaluated for topical relevance and semantic coherence. Fragments that exhibit low contextual consistency—such as embedded promotional messages, cross-domain snippets, or legal disclaimers—are pruned,while all text segments that potentially carry substantive information (e.g., data analyses, case descrip-

tions, or domain-specific terminology) are retained.

Through the integration of *topical relevance estimation* and *semantic coherence evaluation*, this stage enables the large model to act as a global semantic discriminator.

By integrating *topical relevance estimation* and *semantic coherence evaluation*, this stage enables ChatGPT-4o-mini to act as a global semantic discriminator. A schematic illustration and the corresponding formal description are provided in Appendix C. Empirical results (see Section 4.4) show that LLM-driven semantic pruning substantially improves extraction purity and semantic consistency, yielding consistent gains in CEQI, Precision, and F1. While Recall may decrease due to an inevitable *cleaning tax*—where borderline content is discarded in exchange for higher purity—its absolute value remains high, supporting a favorable trade-off for downstream corpus construction.

### 3.5. Open-Source Implementation

To facilitate the construction of high-quality web content datasets, we have prepared a comprehensive open-source release of Webis. The release includes the full implementation of all core components, covering web content extraction, noise filtering, and corpus construction, together with drivers, example configurations, and lightweight test datasets. We also provide detailed documentation to support installation and reproduction.

To improve usability, we additionally implement a locally deployable one-click frontend with a corresponding backend database. This interface enables users to extract diverse webpage content and export structured outputs through a visual workflow.

For the anonymous review period, we omit the public URLs. Upon acceptance (or at public release), we will provide the repository link and a dedicated documentation website that consolidates all resources for Webis.

## 4. Extractor Evaluation

In this section, we first introduce the datasets, baseline methods, and evaluation metrics utilized in our experiments. Subsequently, we present detailed experimental results and analyzes. Our evaluations specifically address three critical research questions (RQs):

- **RQ1:** How effectively does our method predict text noise at the webpage node level?

- **RQ2:** Under a stable threshold configuration, how does our method compare with state-of-the-art approaches in terms of end-to-end extraction quality and execution robustness on real-world webpages?

- **RQ3:** How well does our method perform in semantic-level filtering?

## 4.1. Experimental Setup

### 4.1.1. DATASETS

To ensure rigorous evaluation while keeping the experimental budget tractable, we build two evaluation sets on top of the SWDE corpus: SWDE-huge (1,600 pages) as our primary testbed for end-to-end performance, and SWDE-small (105 pages) for node-level ablations and model-replacement studies that require repeated inference runs.

The original SWDE corpus spans 8 vertical domains. We reproducibly sample SWDE-huge with a two-stage procedure: (i) for each website, we draw a fixed number of pages $k$ (default 20) with a fixed random seed; (ii) we then perform quota-based stratified filling by "domain→site" until reaching 1,600 pages. We further derive a lightweight SWDE-small that keeps 15 pages per domain for 7 domains (105 pages total) for high-iteration ablation studies.

SWDE-huge retains all 8 domains to reflect diverse real-world layouts and boilerplate noise. SWDE-small intentionally excludes the nbaplayer (NBA players) domain and is used only for cost-efficient node-level ablations rather than as the main benchmark.

For text-level evaluation on SWDE, each selected webpage is paired with a reference *main content* text as the ground-truth target; the construction procedure and rationale are provided in Appendix E.

### 4.1.2. BASELINE METHODS

Following the taxonomy discussed in the Related Work section, we select a representative set of widely adopted web content extraction tools as baselines. These methods collectively cover the major paradigms reviewed in prior studies, including low-level HTML parsing, heuristic and rule-based extraction, density-based boilerplate removal, and end-to-end extraction pipelines. Specifically, we compare against *Readability* (Baburov, 2023), *HTMLParser* (Lin & Hu, 2010), *Trafilatura* (Barbaresi, 2021; Roh et al., 2019; Barbaresi, 2019), *Goose3* (Lababidi, 2024), *Justext* (Belica, 2024), *Newspaper3k* (Ou-Yang, 2024), *Crawl4AI* (UncleCode, 2024), *magic-html* (OpenDataLab, 2024), and *ReaderLM-v2* (Wang et al., 2025),

### 4.1.3. EVALUATION METRICS

Evaluation metrics consider both local matching and global semantic dimensions.

- **Local metrics**: Precision, Recall, and F1 score, which measure noise removal effectiveness and main-text completeness.

*Node-level evaluation.* For node-level experiments on the DOM node dataset, the task is formulated as a binary classification problem (0 = clean, 1 = noise). Precision, Recall, and F1 are computed for each class and reported using support-weighted aggregation: $P_w = \sum_{c\in\{0,1\}} \frac{n_c}{N} P_c$, $R_w = \sum_{c\in\{0,1\}} \frac{n_c}{N} R_c$, and $F1_w = \sum_{c\in\{0,1\}} \frac{n_c}{N} F1_c$. The reported $F1_w$ is the weighted average of class-wise F1 scores.

*End-to-end extraction evaluation.* For end-to-end experiments on SWDE, local metrics are computed at the text level by comparing the extracted main content with the reference ground-truth content. In this setting, evaluation is defined over main-text matching only, without an explicit noise class. Precision, Recall, and F1 are computed from text overlap statistics and reported using the standard formulation.

- **Global semantic metrics**: TF–IDF cosine similarity(Ramos et al., 2003), Jaccard similarity(Niwattanakul et al., 2013), and BERTScore(Zhang et al., 2019), which capture overall semantic consistency(Powers, 2020).

- **Composite metric CEQI (Content Extraction Quality Index)**: The overall score is computed by weighted fusion of local and global metrics:

$$\text{CEQI} = \alpha\,\text{Precision} + \beta\,\text{Recall} + \gamma\,\text{F}_1 \\ + \delta\,\frac{\text{TF}-\text{IDF} + \text{Jaccard} + \text{BERTScore}}{3}, \quad (9)$$

where $\alpha = 0.2$, $\beta = 0.2$, $\gamma = 0.2$, and $\delta = 0.4$, allocating 60% weight to extraction accuracy (P/R/F1) and 40% to semantic fidelity (the averaged TF–IDF/Jaccard/BERTScore). The weights are fixed *a priori* (not tuned on SWDE test sets), and CEQI is reported only as an auxiliary summary index rather than an optimization target. A weight-sensitivity check that varies the semantic weight $\delta$ in $[0, 1]$ (with $\alpha = \beta = \gamma = (1-\delta)/3$) shows that the ranking on SWDE-huge remains unchanged; Webis consistently achieves the highest CEQI (Appendix F).

## 4.2. RQ1: Evaluation of Node-Level Prediction Capability

To assess the robustness of node-level noise detection across different foundation models, we conduct a cross-model comparison on the same DOM node dataset constructed from ClueWeb22 under identical prompting and inference settings. Beyond node-level prediction, we plug each model into Webis as the node predictor while keeping the rest of the pipeline (including threshold configuration) unchanged, and run the complete end-to-end extraction on SWDE-small. The node-level and end-to-end results are summarized in Table 1.

*Table 1.* Comparison of Node-Level Prediction and End-to-End Extraction Performance on SWDE-small.

| Methods | Node-Level Prediction | | | End-to-End Extraction (SWDE-small) | | | | | | |
|---|---|---|---|---|---|---|---|---|---|---|
| | $P_w$ | $R_w$ | $F1_w$ | Precision | Recall | F1-Score | TF-IDF | Jaccard | BERTScore-F1 | CEQI |
| GPT-4o | 52.5 | 51.4 | 45.2 | 66.9 | 46.7 | 49.7 | 57.5 | 37.0 | 85.2 | 56.6 |
| Qwen2.5-72B-Instruct | 43.3 | 32.8 | 34.1 | 64.7 | 23.4 | 29.9 | 40.7 | 20.8 | 82.5 | 42.8 |
| DeepSeek-V3 | 49.0 | 49.5 | 43.0 | 59.5 | 26.9 | 31.9 | 41.9 | 21.7 | 81.5 | 43.0 |
| Grok-3 | 53.9 | 51.9 | 48.7 | 64.3 | 40.7 | 44.8 | 52.6 | 32.5 | 84.2 | 52.5 |
| Webis (ours) | **82.4** | **82.2** | **82.2** | **80.6** | **63.9** | **66.3** | **71.8** | **56.1** | **86.9** | **70.8** |

*Table 2.* Comprehensive Comparison of End-to-End Web Content Extraction Performance on SWDE-huge. Success Rate reports the percentage of webpages for which a method successfully produces valid extraction outputs. CEQI is a holistic extraction quality index and is reported independently.

| Category | Method | Success Rate | Accuracy Metrics | | | Similarity Metrics | | | CEQI |
|---|---|---|---|---|---|---|---|---|---|
| | | | Precision | Recall | F1-Score | TF-IDF | Jaccard | BERTScore-F1 | |
| Naive | readability-lxml 0.8.1 | 93.9 | 72.8 | 52.4 | 55.6 | 60.0 | 44.3 | 84.7 | 61.4 |
| | htmlparser 0.0.2 | 97.4 | 50.2 | 40.6 | 35.1 | 43.8 | 26.1 | 83.5 | 45.6 |
| | trafilatura 1.12 | **99.9** | 70.8 | 67.1 | 63.2 | 67.0 | 51.4 | 87.4 | 67.6 |
| | goose3 3.1.19 | 83.9 | 74.1 | 42.6 | 50.0 | 56.7 | 39.8 | 85.6 | 57.6 |
| | justext 3.0.1 | 73.1 | 77.7 | 54.4 | 59.5 | 66.7 | 47.6 | 87.4 | 65.2 |
| | Newspaper3k 4.12.3 | 93.9 | 72.2 | 50.2 | 54.5 | 60.0 | 44.3 | 86.2 | 60.8 |
| | magic-html 0.1.8 | 96.8 | 77.8 | 69.8 | 67.6 | 70.9 | 55.5 | **89.1** | 71.8 |
| | Crawl4AI 0.7.8 | 95.0 | 45.0 | 77.9 | 46.2 | 52.6 | 31.2 | 79.4 | 55.6 |
| | ReaderLM-v2 | 80.4 | 39.7 | 44.7 | 34.1 | 33.2 | 18.8 | 80.9 | 41.4 |
| Our Method | Webis | 99.6 | **78.0** | **72.6** | **71.3** | **74.2** | **58.2** | 87.5 | **73.7** |

**Node-level prediction.** Table 1 shows that off-the-shelf LLM predictors struggle to reliably discriminate structurally deceptive noise from fragmented main content: their node-level Precision and Recall only range from 43.3–53.9 and 32.8–51.9, leading to a low F1 range of 34.1–48.7. In contrast, Webis reaches 82.4/82.2 Precision/Recall with 82.2 F1, indicating that coupling semantic cues with DOM-aware structural features yields substantially stronger and more stable node-level separation.

**End-to-end impact.** The node-level gap consistently propagates to end-to-end extraction quality on SWDE-small. Webis achieves 66.3 end-to-end F1 and 70.8 CEQI, whereas representative LLM-based predictors (e.g., GPT-4o and Grok-3) reach only 49.7/56.6 and 44.8/52.5 in end-to-end F1/CEQI, respectively. This confirms that improving node-level selection is not merely a local gain: it directly reduces noise retention and omission errors, yielding better lexical overlap and global semantic alignment in the final extracted text.

### 4.3. RQ2: Comprehensive Evaluation of Overall Performance

After fixing the threshold configuration for the reverse-coloring stage, we evaluate the end-to-end web content

extraction performance of Webis on SWDE-huge and compare it against existing methods; the threshold selection and sensitivity analysis are provided in Appendix H. Table 2 reports the overall results on SWDE-huge, including extraction robustness (Success Rate), Precision/Recall/F1, and multiple semantic similarity metrics (TF-IDF, Jaccard, BERTScore-F1, and CEQI), providing a multi-dimensional assessment of end-to-end extraction quality.

**Robustness and overall quality.** All extraction quality and semantic similarity metrics are computed on the subset of webpages (out of 1,600 pages) for which each method successfully produces valid, evaluable outputs, ensuring a fair comparison. On this evaluation subset, Webis attains a 99.6% success rate and achieves the best overall CEQI of 73.7, with Precision/Recall/F1 of 78.0/72.6/71.3. This indicates that Webis maintains near-complete coverage of real-world webpages while producing high-quality extracted text.

**Comparison with strong baselines.** Table 2 also reveals a robustness–quality distinction among competitive baselines. For example, Trafilatura achieves the highest Success Rate (99.9%) but its CEQI is 67.6, substantially lower than Webis (73.7), suggesting that robustness alone does not guarantee

*Table 3.* Comparison of Each Method's Performance After Semantic Cleansing and the Corresponding Increments (Including Similarity Metrics).

| Method | Cleaned Acc | | | Δ Acc | | | Cleaned Sim | | | | Δ Sim | | | |
|---|---|---|---|---|---|---|---|---|---|---|---|---|---|---|
| | P | R | F1 | ΔP | ΔR | ΔF1 | TF-IDF | Jaccard | BERT-F1 | CEQI | ΔTF | ΔJa | ΔBF | ΔCQ |
| Goose3 | 74.0 | 36.9 | 45.6 | −0.1 | −5.7 | −4.4 | 53.8 | 36.7 | 73.8 | 53.2 | −2.9 | −3.1 | −11.8 | −4.4 |
| htmlparser | 75.9 | 34.7 | 40.3 | +25.7 | −5.9 | +5.2 | 48.2 | 31.6 | 79.6 | 51.4 | +4.4 | +5.5 | −3.9 | +5.8 |
| Justext | 85.1 | 47.9 | 56.6 | +7.4 | −6.5 | −2.9 | 65.8 | 45.9 | 84.7 | 64.1 | −0.9 | −1.7 | −2.7 | −1.1 |
| Newspaper3k | 75.7 | 44.6 | 51.9 | +3.5 | −5.6 | −2.6 | 58.4 | 42.6 | 78.9 | 58.4 | −1.6 | −1.7 | −7.3 | −2.4 |
| Readability-lxml | 76.3 | 47.2 | 53.7 | +3.5 | −5.2 | −1.9 | 58.7 | 44.1 | 76.6 | 59.4 | −1.3 | −0.2 | −8.1 | −2.0 |
| Trafilatura | **84.7** | 58.8 | 64.2 | +13.9 | −8.3 | +1.0 | 67.9 | 52.9 | 87.1 | 69.3 | +0.9 | +1.5 | −0.3 | +1.7 |
| magic-html | 88.2 | 64.1 | 69.2 | +10.4 | −5.7 | +1.6 | 72.8 | 57.9 | 85.6 | 73.1 | +1.9 | +2.4 | −3.5 | +1.3 |
| Crawl4AI | 66.9 | 69.4 | 62.5 | +21.9 | −8.5 | +16.3 | 64.5 | 46.0 | 78.9 | 65.0 | +11.9 | +14.8 | −0.5 | +9.4 |
| ReaderLM-v2 | 50.7 | 39.7 | 38.9 | +11.0 | −5.0 | +4.8 | 39.0 | 21.4 | 80.4 | 44.6 | +5.8 | +2.6 | −0.5 | +3.2 |
| **Ours** | 78.0 | **72.6** | **71.3** | +24.6 | −11.2 | +10.3 | **74.2** | **58.2** | **87.5** | **73.7** | +8.6 | +12.0 | +2.8 | +7.8 |

semantic fidelity. Among heuristic extractors, magic-html is the strongest competitor in CEQI (71.8), yet Webis still improves CEQI by +1.9 (73.7 vs. 71.8) and achieves higher TF-IDF and Jaccard similarity (74.2/58.2 vs. 70.9/55.5), while remaining competitive on BERTScore-F1 (87.5 vs. 89.1). Overall, Webis improves extraction quality without sacrificing execution stability on large-scale, structurally complex webpages.

**Across-domain generalization.** Beyond the aggregate score, Figure 5 (Appendix G) presents CEQI across SWDE vertical domains. Webis is leading or near-leading on most domains and exhibits a more uniformly high CEQI profile across domains, indicating that the improvement is not dominated by any single domain but generalizes across diverse page layouts and content types.

### 4.4. RQ3 Semantic-Level Filtering Effect

To quantify the contribution of the semantic-level pruning module (Section 3.4), we perform an ablation study by applying the same semantic cleansing stage to each method's extracted text and reporting the post-cleaning metrics together with the corresponding increments in Table 3. The results reveal a consistent *precision–recall trade-off*: semantic filtering often increases Precision by removing semantically weak or redundant fragments, but it may also reduce Recall when borderline yet partially relevant text is discarded. Moreover, improvements are *not uniform* across baselines—some methods even degrade in both accuracy and semantic similarity after cleaning (e.g., Goose3 shows ΔP= −0.1, ΔR= −5.7, and a large drop in BERT-F1 with ΔBF= −11.8 and ΔCQ= −4.4), indicating that semantic pruning is sensitive to the quality and coverage of the upstream candidate text space.

For Webis, semantic-level pruning yields substantial gains in overall quality: Precision increases from the pre-cleaning setting by ΔP= +24.6 to 78.0, while F1 improves by ΔF1= +10.3 to 71.3 (Table 3). Although Recall decreases by ΔR= −11.2 to 72.6, this reduction is expected under

semantic filtering and is also observed in other strong extractors (e.g., Trafilatura with ΔR= −8.3 and Crawl4AI with ΔR= −8.5). Importantly, Webis is the only method that achieves *simultaneous improvements on all semantic metrics*, with TF-IDF (+8.6), Jaccard (+12.0), BERT-F1 (+2.8), and CEQI (+7.8), suggesting that our pruning removes noise without sacrificing global semantic fidelity.

These patterns suggest that semantic pruning is most effective when the upstream stages provide a high-coverage and structurally grounded candidate set. Methods lacking strong structural constraints can exhibit limited or even negative semantic gains after cleaning: for instance, htmlparser obtains a large Precision increase (ΔP= +25.7) but still suffers a notable BERT-F1 drop (ΔBF= −3.9), and several heuristic pipelines show negative increments on multiple similarity metrics (e.g., Justext and Readability-lxml). In contrast, Webis couples high-recall structural filtering with global semantic discrimination, enabling the semantic stage to act as a *refinement* step rather than a destructive filter, thereby improving both content purity and semantic alignment in a coordinated manner.

## 5. Conclusion

This paper addresses the persistent challenge of extracting clean, semantically coherent content from noisy and structurally diverse web pages—a problem that directly impacts corpus quality for downstream NLP applications, including large language model pretraining.We propose **Webis**, a three-stage extraction framework that integrates structure-aware node classification, subtree-level reverse-coloring, and LLM-guided semantic pruning. This design allows Webis to handle both shallow layout noise and deep semantic contamination in a unified and efficient manner. Empirical results across large-scale web benchmarks demonstrate that Webis outperforms existing extraction methods in content purity, structural integrity, and semantic completeness, while remaining adaptable to diverse real-world deployment scenarios.

## Impact Statements

High-quality, low-noise data processing is a critical foundation for both training modern AI systems and enabling reliable downstream deployment. Yet, such work is often engineering-intensive, tedious, and comparatively underappreciated. This paper focuses on web content extraction—a prototypical high-noise setting—and proposes a more robust extraction approach that reduces both structural and semantic noise while preserving the key information in the main content. We aim to provide reusable methods and practical guidance for data acquisition and cleaning in machine learning, ultimately improving the quality and usability of data pipelines for real-world AI systems. Overall, our work is primarily geared toward strengthening data processing capabilities and is not expected to cause direct negative societal impacts.

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

## A. Noise-Confidence Computation Details

---

**Algorithm 3** Tag Noise Probability Calculation

---

**Require:** Dataset $D = \{(x_i, y_i)\}$, HTML tag set $\mathcal{T}$
**Ensure:** Probability dictionary $P = \{t : (p_t, \text{type})\}$
1: Initialize counters: $\forall t \in \mathcal{T}$, $n_t^{\text{noise}} \leftarrow 0$, $n_t \leftarrow 0$
2: **for** each sample $(x_i, y_i) \in D$ **do**
3:     Extract tags $\mathcal{T}_i \leftarrow \text{extract\_tags}(x_i)$
4:     **for** each tag $t \in \mathcal{T}_i$ **do**
5:         $n_t \leftarrow n_t + 1$
6:         $n_t^{\text{noise}} \leftarrow n_t^{\text{noise}} + \mathbb{I}(y_i = 1)$
7:     **end for**
8: **end for**
9: $\text{total\_count} \leftarrow \sum_t n_t$
10: $\alpha_g \leftarrow \lfloor 0.001 \times \text{total\_count} \rfloor$
11: $\beta_g \leftarrow 2 \times \alpha_g$
12: $\pi_g \leftarrow \frac{(\sum_t n_t^{\text{noise}}) + \alpha_g}{(\sum_t n_t) + \alpha_g + \beta_g}$
13: $\text{s} \leftarrow \log_2(|\mathcal{T}|)$
14: $\text{counts} \leftarrow \{n_t \mid t \in \mathcal{T}\}$
15: $\text{low\_freq\_thresh} \leftarrow \text{percentile}(\text{counts}, 10)$
16: **for** each tag $t \in \mathcal{T}$ **do**
17:     **if** $n_t < \text{low\_freq\_thresh}$ **then**
18:         $\pi_{\text{local}} \leftarrow \frac{n_t^{\text{noise}} + s}{n_t + 2 \times s}$
19:         $p_t \leftarrow 0.3 \times \pi_{\text{local}} + 0.7 \times \pi_g$
20:     **else**
21:         $p_t \leftarrow \frac{n_t^{\text{noise}} + s}{n_t + 2 \times s}$
22:     **end if**
23:     $P[t] \leftarrow p_t$
24: **end for**

---

**Baseline Confidence.** Given a node path $P$, let:

- $p_\ell$: noise risk probability of the terminal (leaf) tag;

- $\mathcal{R}$: set of unique tags along the path;

- $\mathcal{R}' = \mathcal{R} \setminus \{\ell\}$: non-terminal tags;

- $N_o = |\mathcal{R}'|$: number of non-terminal tags;

- $w_r, w_o$: weights for terminal and non-terminal tags, respectively.

We establish a baseline confidence score by defining risk Aggregate $M_b$ and effective weight $D_b$:

$$
\begin{aligned}
M_b &= w_r \, p_\ell + w_o \sum_{\tau \in \mathcal{R}'} p_\tau, \\
D_b &= w_r + w_o N_o, \\
C_b &= \frac{M_b}{D_b}.
\end{aligned}
\tag{10}
$$

This formula integrates both terminal and non-terminal tags, thereby balancing the contributions of node-specific and structural risks.

**Dynamic Penalty Mechanisms.** To handle cases with insufficient or excessive tag diversity, we introduce mutually exclusive penalty adjustments:

1. *Single-tag Penalty ($N_o = 0$)*: Applied to mitigate overfitting when only a single tag is present along the path.

$$
\Delta M_\alpha = \frac{\alpha}{2}, \quad \Delta D_\alpha = \alpha \tag{11}
$$

Here, $\alpha$ is the penalty strength parameter.

2. *Multi-tag Depth-sensitive Penalty ($N_o \geq 1$)*: Addresses structural complexity through a depth-based exponential decay function:

$$
\Delta M_\beta = \beta \, \lambda^{\min(d,50)/5}, \quad \Delta D_\beta = \beta \tag{12}
$$

Here, $\beta$ is the diversity penalty factor, $0 < \lambda < 1$ is the depth-decay rate, and $d$ is the DOM node depth, capped at 50 to control complexity.

**Final Noise-Confidence Score.** Combining baseline confidence and dynamic penalties, the noise-confidence metric is computed as:

$$
\begin{aligned}
M &= M_b + \delta_{N_o=0}\Delta M_\alpha + \delta_{N_o \geq 1}\Delta M_\beta, \\
D &= D_b + \delta_{N_o=0}\Delta D_\alpha + \delta_{N_o \geq 1}\Delta D_\beta, \\
C &= \text{clip}\left(\frac{M}{D}, \, 0, \, 1\right),
\end{aligned}
\tag{13}
$$

where $\delta_{\text{cond}}$ is an indicator function (equals 1 if the condition is true; otherwise, 0). The $\text{clip}(x, 0, 1)$ function ensures the final score is bounded within the valid probability range.

## B. Node-Level Feature Encoding Details

We formalize the node representation as the concatenation of multiple feature groups. Let $v_i$ denote the $i$-th DOM node and $\phi(\cdot)$ the overall feature-extraction function decomposed into $m$ sub-functions $\{\phi_j(\cdot)\}_{j=1}^m$. The node feature vector is:

$$\mathbf{x}_i = \phi(v_i) = \big[\phi_1(v_i)\big] \oplus \big[\phi_2(v_i)\big] \oplus \cdots \oplus \big[\phi_m(v_i)\big], \quad (14)$$

where $\oplus$ denotes concatenation.

In our implementation, the core features can be summarized as:

$$\mathbf{x}_i = \Big[ \text{content}(v_i), \ \text{lastTag}(v_i), \ [\text{riskTag}(v_i)], \\ \text{depth}(v_i), \ \text{confidence}(v_i) \Big]. \quad (15)$$

where $\text{riskTag}(v_i)$ is the discretized tag-level prior (Appendix **??**) and $\text{confidence}(v_i)$ is the path-aware noise-confidence score (Appendix A).

## C. Page Noise Cleaning Illustration

This appendix complements the main text by providing a visual and formal description of the page noise cleaning process, with emphasis on the Semantic-Level Pruning stage (see Section 3.4) where prompt-optimized LLM cleaning is applied to the text produced by preceding stages to remove Level-2 noise.

**Prompt-Optimized LLM Cleaning: Formal Description**

This subsection provides a formal description of the prompt-optimized LLM cleaning procedure used in the Semantic-Level Pruning stage. Note that this step does not re-parse the DOM tree; rather, it refines the textual output produced by prior structured preprocessing and reverse-coloring.

Let the set of candidate textual segments obtained after structural preprocessing and reverse-coloring be

$$S = \{s_1, s_2, \ldots, s_n\},$$

or let $D = \text{Concat}(S)$ denote the concatenated document. We perform prompt-optimized cleaning by issuing a prompt $P$ to the LLM $f_\theta$ (e.g., DeepSeek-V3) and applying the model to the extracted text.

We consider two interchangeable modes of operation: paragraph-level cleaning and whole-document cleaning. In the paragraph-level mode, each segment $s_i$ is processed together with page context $C$; the model returns a cleaned segment $\tilde{s}_i$ and a relevance score $r_i$:

$$(\tilde{s}_i, r_i) = f_\theta^P(s_i \mid C), \qquad i = 1, \ldots, n,$$

where $f_\theta^P$ denotes the prompt-guided model invocation and $r_i \in [0,1]$ quantifies the segment's topical relevance as judged by the model (this score can be produced directly by the LLM or derived from model logits / calibrated outputs).

Alternatively, in the whole-document mode the model may directly output a cleaned document $D^*$ and, optionally, a set of per-segment relevance scores $\{r_i\}$:

$$\big(D^*, \{r_i\}_{i=1}^n\big) = f_\theta^P(D).$$

For the paragraph-score-based selection, a global relevance threshold $\tau$ is applied and only cleaned segments with $r_i \geq \tau$ are retained and concatenated in the original order to form the final output:

$$S^* = \{\, \tilde{s}_i \mid r_i \geq \tau, \ i = 1, \ldots, n \,\}, \\ D^* = \text{Concat}\big(S^*\big).$$

Here, $\tau$ is a tunable parameter (set by validation or downstream objectives). The score $r_i$ may be produced as a direct probability, a normalized score, or as the confidence associated with an LLM-produced "On-Topic / Off-Topic" label.

For reproducibility, a simplified example of a paragraph-level cleaning prompt is provided below. This prompt guides the model to judge whether a paragraph $s_i$ is on-topic given the page context $C$, and to return either a cleaned paragraph or an explicit off-topic marker.

*Listing 1.* Text Refinement Prompt Example

```
You are a recall-oriented content cleaner.
You receive the visible plain text of a web page.
Your job is to delete ONLY content that is clearly
    boilerplate or clearly unrelated.
When uncertain, KEEP the text.
 KEEP (prefer keeping borderline content):
- Main descriptive content about the entity/topic
- Reviews, personal experiences, quotes, narratives
- Semi-structured factual fields and short lines (e.
    g., address, phone, hours, price, tags,
    categories, ISBN, cast, credits)
- Lists that describe the topic (features,
    ingredients, tracklist, filmography, menu items,
    stats) if they appear relevant
 REMOVE (only when clearly boilerplate):
- Navigation/UI chrome: repeated site-wide menus,
    headers/footers, breadcrumbs
- Login/signup, cookie notices, legal/copyright, ads
    /sponsored blocks
- Pure link directories unrelated to the topic (e.g
    ., site-wide city lists) and repeated template
    text
Rules:
- Do NOT rewrite, summarize, translate, or add new
    text.
- Preserve the original order and formatting; keep
    line breaks.
```

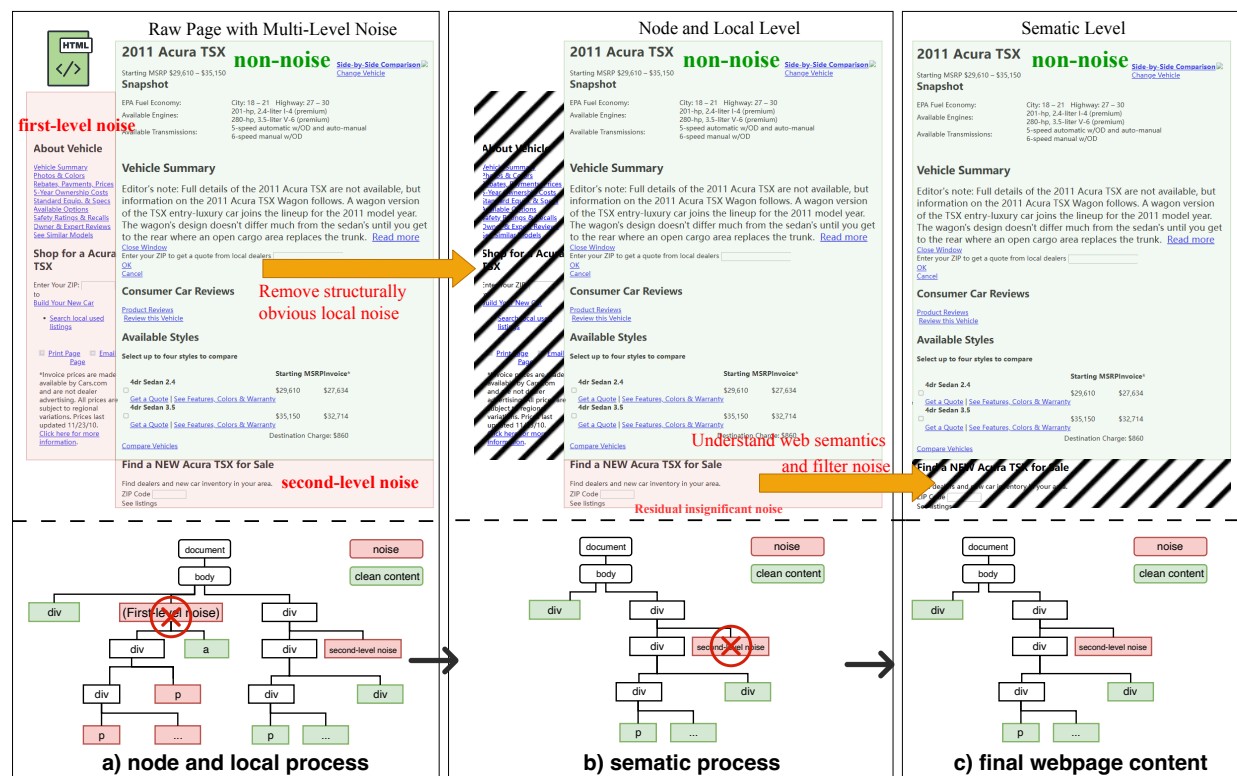

*Figure 3.* Illustration of the page noise cleaning process. Multi-level noise is categorized into *Level-1* (structurally identifiable noise) and *Level-2* (structurally plausible but semantically off-topic). Node- and subtree-level processing (Sections 3.2–3.3) primarily remove Level-1 noise; semantic-level filtering (Section 3.4) applies prompt-optimized LLM cleaning to the candidate text to remove Level-2 noise.

```
- If nothing relevant exists, output an empty string
  .
Text:
```

In practice, this prompt-optimized cleaning procedure is used to remove Level-2 noise (semantically off-topic but structurally plausible fragments). The quantitative impact of the procedure—e.g., improvements in CEQI and Precision/Recall—is reported in Section 4.4.

## D. Implementation Details

### D.1. Dataset Construction

To ensure both DOM-Validator and the node-level noise classifier are trained on diverse, bias-minimized samples, we curate a large-scale dataset from ClueWeb22(Overwijk et al., 2022) as follows:

- **Page Sampling:** Randomly sample 40,000 web pages and select those with complex structure (XPath depth $\geq 5$) and moderate text density (500–5,000 characters).

- **Node Filtering:** Parse each HTML with BeautifulSoup (e.g. `lxml`) into a DOM tree. Perform DFS traver-

sal—skipping `<script>`, `<style>`, and `<meta>`—to extract every text-bearing node. Compute each node's tag path (e.g. `div[2]/p[1]`) by indexing sibling order, and record tag name, attributes, depth, path, and content.

- **Quality Control:** Discard nodes with text length $< 10$ or entropy $< 2$. This yields a balanced dataset of positive (noise) and negative (content) samples.

- **Dataset Partitioning:** We assign 36,000 web pages—approximately 84M DOM nodes—to the training set and 4,000 pages—approximately 9M nodes—to the test set, ensuring a balanced 1:1 ratio between noise and content nodes.

### D.2. DOM-Validator Statistical Learning

DOM-Validator statically analyzes each page to extract structural and semantic features relevant to noise detection and risk assessment, then applies statistical learning to estimate tag-level risk probabilities and node confidence scores.

- **Risk-Tag Generation:** As described in Section 3.2.2, we traverse all annotated nodes in the training set to collect the frequency and noise-rate of each DOM

tag (e.g. `<div>`, `<aside>`). This yields mappings ⟨tag, noise probability⟩ that serve as priors for new pages.

- **Noise-Confidence Estimation:** Combining tag-level probabilities with node depth, we compute a normalized confidence score via weighted fusion and Bayesian smoothing. An adaptive depth-decay mechanism and global–local weighting jointly ensure robust estimates across depths and tag frequencies.(Sun et al., 2024)

### D.3. Node-Level Noise Prediction Model Fine-Tuning

To refine local noise judgments, we fine-tune the Qwen2.5–1.5B model using the structured and semantic features produced by DOM-Validator. Each training example concatenates risk tags, noise probability, and text content into a standardized prompt. This enables the model to jointly learn hierarchical risk signals (e.g. `<aside>[H]`) and semantic cues (e.g. "limited-time offer").

To prevent overfitting, we employ a custom LoRA(Hu et al., 2022) adapter with rank 16 and scaling factor $\alpha = 32$, applied to the attention query/value projections, disabling rank stabilization and decomposition extensions. A sparse gating mechanism restricts 30% of neurons per layer for local feature reconstruction, and a 0.2 DropPath further regularizes the parameter space. This dual-regularization strategy yields a compact yet robust classifier for node-level noise detection.

## E. Construction of SWDE "True Content"True Content References

To enable text-level evaluation on SWDE (e.g., Precision/Recall/F1 and semantic similarity), we construct a per-page reference text, termed *true content*, as the ground-truth target paired with each sampled webpage in SWDE-huge and SWDE-small (Section 4.1.1).

**Step 1: Visible-text extraction.** For each SWDE webpage, we first extract its human-visible text by removing scripts/styles and collecting rendered textual blocks in reading order, yielding a plain-text input that is independent of any specific extractor implementation.

**Step 2: LLM-based true-content generation (GPT-5.2).** We then prompt GPT-5.2 to extract the main content from the visible plain text under a recall-oriented instruction: remove only clear boilerplate/template content while preserving all substantive paragraphs and semi-structured factual fields. The model is required to *not* rewrite, summarize, translate, or introduce new information, and to keep the original order and line breaks whenever applicable. The resulting output is used as the initial *true content* reference

$R$.

**Step 3: Human sampling audit and correction.** To improve trustworthiness, we conduct human sampling audits across domains and websites. Auditors compare the generated reference against the rendered webpage to check for (i) major omissions of topic-relevant content and (ii) inclusion of clearly irrelevant boilerplate or hallucinated text. When violations are found, we correct the reference by re-running extraction with a stricter constraint or performing manual edits following the same inclusion/exclusion guideline.

**Ground-truth rationale and consistency with the main text.** This procedure makes the reference grounded in human-visible evidence (Step 1), standardized by a single, consistent extraction policy (Step 2), and partially validated by humans (Step 3). Therefore, the resulting *true content* serves as a reasonable and method-agnostic ground-truth target for comparing different extractors under the same evaluation protocol. All local and semantic metrics reported in Section 4.1.3 and RQ2/RQ3 are computed by comparing each method's output $E$ with the corresponding reference $R$.

## F. CEQI Weight Rationale and Sensitivity

CEQI is introduced as a compact reference index rather than an optimization target. We fix the default weights $(\alpha, \beta, \gamma, \delta) = (0.2, 0.2, 0.2, 0.4)$ *a priori* to balance basic accuracy (Precision/Recall/F1) and semantic fidelity (TF–IDF, Jaccard, and BERTScore), and we do not tune these weights on SWDE test sets.

## G. Additional Results on SWDE Vertical Domains

**Sensitivity protocol.** We assess robustness by varying the local–semantic trade-off through a single parameter $\delta$:

$$\alpha = \beta = \gamma = \frac{1-\delta}{3}, \quad \delta \in [0, 1],$$

while keeping the internal local weights equal. Using the already reported component metrics in Table 2, we recompute CEQI values for each $\delta$ without rerunning any extraction experiments. Since CEQI is linear in $\delta$, ranking changes can be observed directly by interpolating between the two extremes: $\delta = 0$ (only local accuracy) and $\delta = 1$ (only semantic fidelity).

## H. Threshold Selection and Sensitivity Analysis

**Result.** Across all $\delta \in [0, 1]$, the method ranking on SWDE-huge remains unchanged. Webis is the top-

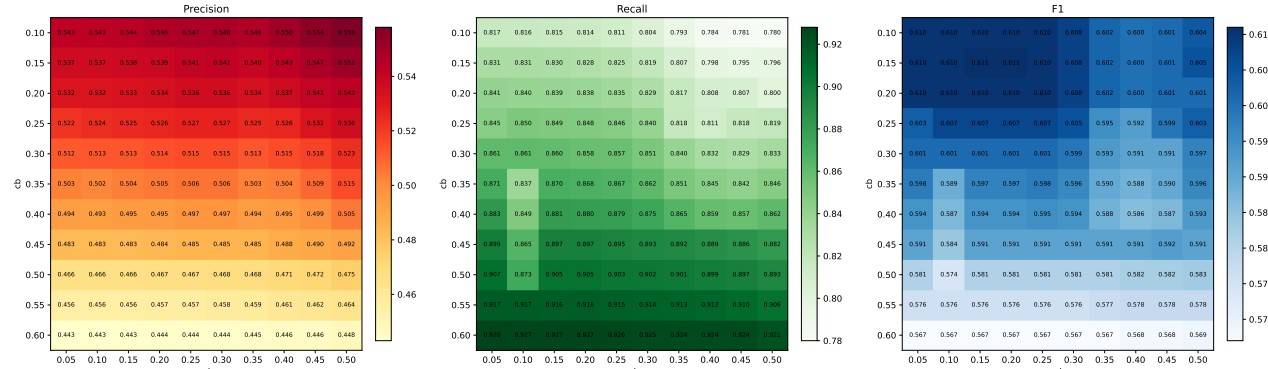

*Figure 4.* Heatmap analysis of reverse-coloring thresholds on SWDE-huge. Left: Precision; Middle: Recall; Right: F1-Score. The horizontal axis denotes the whitening threshold $T_1$, and the vertical axis denotes the pruning threshold $T_2$. A broad high-performance plateau is observed for F1-Score, indicating robust and stable behavior under moderate threshold settings.

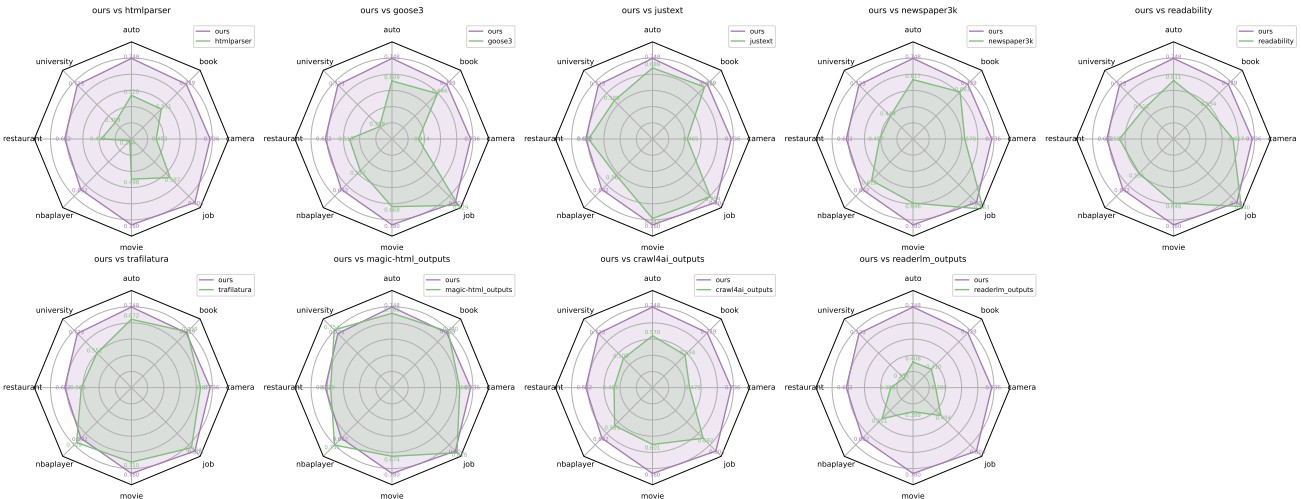

*Figure 5.* Comparison of CEQI scores across vertical domains in SWDE: Our method is compared against baseline approaches across seven domains—auto, book, camera, job, movie, restaurant, and university—demonstrating its effectiveness in comprehensive content extraction quality.

performing approach at both endpoints and throughout the interpolation, showing that the relative conclusion is not driven by a specific weight selection. Small random perturbations around the default weights (within $\pm 0.1$ followed by normalization) also preserve the same ordering.

**Interpretation.** Therefore, CEQI serves as an auxiliary and stable summary index, reflecting a balanced view of extraction accuracy and semantic fidelity. All component metrics are still reported in the main text, ensuring transparency and preventing potential manipulation by arbitrary weighting.

Before the end-to-end evaluation, we conduct an exploratory analysis of the key thresholds in the reverse-coloring stage to verify stable and controllable behavior within reason-

able ranges and to assess whether performance depends on fragile threshold choices.

Reverse-coloring makes subtree-level global decisions using two thresholds: the whitening threshold $T_1$ determines whether a subtree is globally marked as clean, and the pruning threshold $T_2$ decides whether a high-noise subtree is removed; this design must balance deep-noise suppression against accidental removal of large shallow subtrees.

We systematically vary $T_1$ and $T_2$ on SWDE-huge while retaining only node-level noise prediction and reverse-coloring to isolate threshold effects; Figure 4 shows heatmaps of Precision, Recall, and F1-Score, where increasing $T_2$ yields a typical trade-off of higher Recall and lower Precision, and F1 forms a high-value plateau over moderate ranges, indicating robustness rather than reliance on an

isolated optimum.

Accordingly, we adopt $T_1 = 0.2$ and $T_2 = 0.2$ as the default setting, which lies near the center of the plateau and is insensitive to moderate perturbations, yielding a stable balance between Precision and Recall.

Figure 5 visualizes CEQI comparisons between Webis and multiple baselines across SWDE vertical domains using radar charts, serving as a cross-domain check of consistency and generalization.

Overall, Webis expands outward beyond competing methods on most domain axes, exhibiting leading or near-leading CEQI, which suggests that its improvements are not dominated by any single domain but remain stable across domains.

Meanwhile, some strong baselines become competitive in certain domains, yet Webis maintains a more uniformly high profile across domains, reflecting stronger robustness and adaptability to diverse webpage structures and content types.

