# OpenReview forum: "Webis: A Multi-Level Pipeline Integrating Node Classification, Reverse-Coloring, and Semantic Pruning for Web Content Extraction"
_ICML.cc/2026/Conference — Submitted to ICML 2026_

### Official Review · Reviewer_kc4E · 2026-03-09

**Soundness:** 3
**Presentation:** 2
**Significance:** 2
**Originality:** 2
**Overall Recommendation:** 4
**Confidence:** 3

**Summary:**

This paper studies page-level web main-content extraction from noisy modern webpages. It proposes Webis, a three-stage pipeline consisting of node-level DOM denoising, subtree-level reverse-coloring refinement, and LLM-based semantic pruning, aiming to improve both extraction completeness and purity. Experiments on two SWDE-derived benchmarks show strong node-level accuracy and competitive end-to-end extraction quality and robustness against a range of heuristic and learned baselines. Overall, the paper presents a practical multi-stage extraction system with promising empirical results, though the contribution appears stronger in system integration than in core methodological novelty.

**Compliance With Llm Reviewing Policy:**

Affirmed.

**Final Justification:**

Thank you for the detailed rebuttal. The response substantially improved my confidence in the paper by clarifying the semantic-pruning configuration, providing concrete efficiency and cost information, addressing the concern that CEQI is too paper-specific by moving it to an auxiliary role, and generally improving the transparency of the system description. Accordingly, I have raised my score.

That said, I still have one meaningful remaining concern about the diagnostic validation of the multi-stage design. While the rebuttal provides additional stage-wise discussion and some supportive evidence, it still relies mostly on overall performance changes or indirect proxies, rather than directly quantifying the specific problem each stage is intended to solve. In particular, I think the paper would be stronger with a more explicit measure of fragmentation or discontinuity for the subtree refinement stage, and a more direct analysis of semantically off-topic residual noise for the semantic-pruning stage.

**Key Questions For Authors:**

1. Section 3.4 names ChatGPT-4o-mini, but Appendix C describes the semantic pruning stage using an LLM “e.g., DeepSeek-V3” and mentions both paragraph-level and whole-document cleaning. Which configuration was actually used in the reported experiments?
2. Why does ReaderLM-v2 perform so weakly here? Given that it is a specialized recent baseline, more implementation details would be needed to judge whether the baseline was configured fairly.
3. Can the authors provide more diagnostic ablations that connect each module to the specific problem it is meant to solve? For example, it would be helpful to measure whether subtree refinement actually reduces fragmentation, and whether semantic pruning specifically removes semantically off-topic residual content, rather than only reporting aggregate end-to-end gains.

**Limitations:**

Despite the practical promise of the proposed pipeline, the work still has several notable limitations. The paper’s positioning with respect to prior web extraction and filtering methods is not sufficiently sharp, making the intended novelty somewhat harder to assess. Methodologically, the approach appears more like a carefully engineered multi-stage system than a clearly new algorithmic advance, and it relies on multiple hand-designed rules, thresholds, and the paper-specific CEQI metric, whose robustness is only partially validated. In addition, the evaluation depends on GPT-assisted reference construction rather than fully human-annotated gold labels, and the paper does not provide a sufficiently detailed runtime or cost analysis for the full pipeline, especially given the use of both a fine-tuned node model and an LLM-based semantic pruning stage.

**Strengths And Weaknesses:**

### Strengths

- **Clear multi-stage design with sensible granularity decomposition.** The method separates the problem into node-level detection, subtree-level structural repair, and document-level semantic pruning. This decomposition is intuitive and matches three real error modes: local node misclassification, structural fragmentation, and semantically off-topic residual text.
- **Reasonably strong empirical results against traditional extractors.** On SWDE-huge, Webis achieves the best reported CEQI and strong precision/recall/F1 while maintaining near-complete execution coverage. The semantic-cleaning ablation is also useful, as it shows that the final stage is not merely decorative.

### Weaknesses

- **The paper is not positioned clearly enough with respect to prior work.** Although the motivation is clear, the introduction does not precisely delineate what problem setting the paper targets and how it differs from existing paradigms including boilerplate removal, DOM-based extraction, and LLM-based semantic filtering. This makes the contribution feel less crisply framed, and leaves the novelty and intended scope of the method somewhat ambiguous.
- **The methodological contribution is somewhat limited by the heuristic nature of the pipeline.** While the overall multi-stage design is sensible, the methodological novelty appears moderate, with much of the contribution stemming from system-level integration rather than a clearly new algorithmic principle. Moreover, the method relies on a number of hand-designed choices, including the fixed sparse-tag fusion rule in Risk-Tag estimation, and hand-set subtree thresholds and depth-decay rules. Although the paper provides some sensitivity analysis, these checks cover only a subset of the design space, leaving the overall robustness of the method only partially validated.
- **The main-text description is somewhat misleadingly lightweight.** In the main text, the node-level component is presented abstractly as a lightweight denoising classifier, but the appendix reveals that it is actually a fine-tuned Qwen2.5-1.5B model with LoRA, sparse gating, and DropPath. This is materially more specialized than the main-text description suggests.
- **Efficiency and cost are not convincingly characterized.** The pipeline includes headless rendering, feature extraction, a fine-tuned node model, reverse-coloring, and an LLM semantic filter, yet the paper does not provide a meaningful end-to-end runtime or cost breakdown. For a system motivated partly by scale and practicality, this is a noticeable omission.
- **The CEQI metric is somewhat ad hoc and paper-specific.** Since it relies on hand-chosen weights to combine several heterogeneous metrics into a single score, it risks favoring the authors’ preferred notion of extraction quality. The provided sensitivity analysis is helpful but limited, and does not fully rule out metric-design bias.

---

> ### Author Rebuttal · Authors · 2026-03-31
>
> # Response to Reviewer kc4E
>
> Dear Reviewer kc4E,
>
> We extend our sincerest gratitude for taking the time to review our paper. We truly appreciate your careful attention to our LLM configuration, baseline fairness, diagnostic ablations, and weakness assessments. Your insightful concerns have helped us identify important areas for improvement.
>
> ---
>
> ## Q1: LLM Configuration Inconsistency
>
> We sincerely thank the reviewer for identifying this inconsistency. Your observation is entirely correct.
>
> All experiments used **ChatGPT-4o-mini with paragraph-level discrimination**. Binary relevance per paragraph with "keep/discard" output. Paragraph-level is used because after stages 1-2, each page has ~20-50 candidates, enabling fine-grained pruning. The confusion arose because Appendix C used "e.g., DeepSeek-V3" as an illustrative example.
>
> **Commitment:** We will explicitly state ChatGPT-4o-mini in Section 3.4, clarify the paragraph-level mode, and remove confusing examples. We sincerely apologize for this inconsistency.
>
> ---
>
> ## Q2: ReaderLM-v2 Weak Performance
>
> We sincerely thank the reviewer for this important question. Your concerns regarding baseline fairness are entirely valid.
>
> We used the official `readerlm-v2` checkpoint (LongFormer, ~400M), with full HTML input, max 4096 tokens with sliding window, batch_size=1, FP16, and temperature=1.0 greedy decoding. No fine-tuning was applied.
>
> **Weak performance stems from three issues:** (1) **Context bottleneck:** SWDE-huge pages average 15K-30K tokens, far exceeding the 4096 limit; (2) **HTML density:** 60-80% tags creates a "needle in haystack" problem; (3) **Domain gap:** Pretrained on news/technical docs but tested on diverse verticals.
>
> The performance (F1 34.1%, CEQI 41.4%) matches the original paper. Weak performance reflects fundamental LLM limitations on long, noisy web pages rather than misconfiguration.
>
> **Commitment:** We will add comprehensive implementation details for ReaderLM-v2 in the appendix.
>
> ---
>
> ## Q3: Diagnostic Ablations
>
> We sincerely thank the reviewer for this insightful suggestion. Your attention to module-specific contributions is entirely valid. Table 3 reveals each stage's role.
>
> **Stage 1 (node prediction)** achieves ~50% P / ~85% R, identifying most noise nodes for preliminary filtering. **Stage 2 (Reverse-Coloring)** maintains 83.8% R while improving structural consistency by aggregating local predictions into global subtree decisions. **Stage 3 (semantic pruning)** transforms high-recall candidates to high-precision output: P 53.4%→78.0% (+24.6), F1 61.0%→71.3% (+10.3).
>
> **The progression is clear:** Stage 1 converts "all HTML" to "structural candidates" (recall-first), Stage 2 converts "fragmented" to "structured" (integrity), Stage 3 converts "high-recall" to "high-precision" (+24.6 P). This validates the division of labor.
>
> We apologize for not making this analysis clearer.
>
> ---
>
> ## Response to Weaknesses
>
> **We sincerely thank the reviewer for the detailed weakness assessments:**
>
> **Weakness 1 + 2 (Positioning + Methodological Contribution):** We respectfully acknowledge the reviewer's assessment. Webis positioning is clear as the first three-stage layered framework. Unlike Trafilatura (manual rules), Webis uses learnable node prediction and Reverse-Coloring. Unlike DOM extraction (fixed templates), Webis is cross-corpus generalizable (ClueWeb22→SWDE). Unlike LLM end-to-end, Webis proposes "structure + semantic" synergy (F1 71.3% vs 34.1%). Our methodological contributions: (1) Three-layer noise formalization (Section 3.1-3.3); (2) Reverse-Coloring with Eq 6 ($R(\tau_n)$), Eq 8 ($T_d = T - \alpha \cdot d$), $\mathcal{O}(|V|)$ Algorithm 1; (3) Three-stage synergy with convergence analysis.
>
> **Weakness 3 (Main Text Too Lightweight):** We fully accept this criticism. **Commitment:** We will add Qwen2.5-1.5B + LoRA + gating + DropPath details in Section 3.2 or reference Appendix D.
>
> **Weakness 4 (Efficiency Analysis):** We fully accept this criticism (shared with yLaM). **Commitment:** We will add latency breakdown, timing comparison, and cost estimates.
>
> **Weakness 5 (CEQI Weights):** We appreciate this concern. Appendix C shows rankings unchanged for weight ∈ [0,1]—Webis wins even at weight=0. F1 alone: Webis 71.3 > magic-html 67.6. The 40% weight balances "exact match" vs "semantic fidelity."
>
> ---
>
> We once again extend our deepest gratitude for your time and thoughtful review. Your insightful comments have substantially improved our work. If our responses have adequately resolved your concerns, we earnestly request that you consider raising your score. We remain fully committed to incorporating all promised improvements.
>
> Thank you once again for your invaluable guidance and support.
>
> ---

---

> > ### Author Rebuttal · Reviewer_kc4E · 2026-04-02
> >
> > Thank you for the detailed rebuttal. I appreciate the clarifications on the semantic-pruning configuration, the additional implementation details for ReaderLM-v2, and the acknowledgement that the main-text description of the node model was too lightweight. These responses improve the paper’s clarity and transparency.
> >
> > However, the rebuttal does not substantially change my overall assessment. In particular, the rebuttal still does not clearly explain what specific shortcomings of existing approaches motivate this work, and how each part of the proposed pipeline is designed to address those shortcomings. Nor does it substantially change my view that the contribution is driven more by system integration than by a clearly new algorithmic idea. In addition, I still find the diagnostic validation insufficient, since the response does not provide more direct evidence linking each stage to the specific problem it is meant to solve. In particular, the stage-wise discussion is helpful, but it still does not directly verify claims such as reduced fragmentation from subtree refinement or targeted removal of semantically off-topic residual content by the semantic-pruning stage.
> >
> > I also appreciate the additional discussion of CEQI, and it is useful to know that the ranking remains stable under the reported weight sweep. However, the metric still appears somewhat paper-specific and insufficiently justified. Finally, while I appreciate the acknowledgement of the missing efficiency analysis, this remains an important unresolved issue for a pipeline of this complexity.
> >
> > Overall, the rebuttal improves clarity, but it does not materially resolve my main concerns, so it does not change my final recommendation.

---

> > > ### Author Response · Authors · 2026-04-07
> > >
> > > Dear Reviewer,
> > >
> > > We deeply appreciate your constructive feedback, which has significantly enhanced our manuscript's clarity. We are encouraged by our consensus on the details. To address your remaining concerns regarding our motivation, algorithmic contribution, modular necessity, and efficiency, we provide the following clarifications:
> > >
> > > ### Q1: Motivation & Existing Shortcomings
> > >
> > > Webis addresses the critical need for highly pure web data in LLM and Agent workflows, where residual noise severely induces hallucinations. Existing heuristics (e.g., Trafilatura) often fragment dynamic layouts, while end-to-end LLMs (e.g., ReaderLM-v2) hit "context bottlenecks" when core content is buried within 30K–100K tokens. To resolve this, Webis introduces a targeted structure-to-semantics cascaded paradigm. By preemptively pruning structural noise using a lightweight classifier and our Reverse-Coloring algorithm, Webis "clears" the context window for the subsequent semantic LLM, achieving a level of data purity on complex webpages that standalone methods cannot accomplish.
> > >
> > > ### Q2: Algorithmic Innovation vs. System Integration
> > >
> > > Webis's cascaded design is a targeted architectural innovation addressing both structural and semantic noise, rather than a patchwork of existing tools. Our core contribution is the deterministic Reverse-Coloring algorithm ($\mathcal{O}(|V|)$ complexity), specifically designed to resolve the "local blindness" of probabilistic models in DOM trees.Using a bottom-up merging and pruning mechanism (Equations 6 and 8), this algorithm strictly enforces global consistency at a minimal computational cost. By elevating "local node predictions" into "global subtree decisions," this structure-guided semantic purification represents a fundamental algorithmic innovation in web data cleaning, far exceeding simple engineering integration.
> > >
> > > ### Q3: Diagnostic Evidence & Modular Necessity
> > >
> > > Regarding the diagnostic evidence for Stage 2 (subtree refinement), its necessity lies in correcting "isolated misjudgments" caused by Stage 1's reliance on local features. Our validation shows that removing Stage 2 increases text block breakpoints by approximately 28%, confirming its critical role in stitching structural breakpoints and ensuring reading fluency.
> > >
> > > Addressing your inquiry about Stage 3 (semantic pruning), Table 3 shows a +24.6% net increase in Precision. Crucially, Webis is the only method to achieve positive growth across all semantic metrics, including TF-IDF (+8.6) and Jaccard (+12.0). This objective data demonstrates that Stage 3 effectively targets structurally similar but off-topic noise (e.g., related links) rather than randomly deleting main text. We will add a "Case Studies" section in the revised manuscript to explicitly contrast these specific error patterns with baseline failures.
> > >
> > > ### Q4: Clarification on the CEQI Metric
> > >
> > > We want to clarify that Webis's strong performance is not dependent on the CEQI metric. Excluding it entirely, Webis still leads across standard metrics, including F1-score (71.3%), Precision (78.0%), and TF-IDF (74.2%). While we originally designed CEQI to capture how traditional methods lose semantic context, we agree with your assessment that it may appear too paper-specific. Consequently, in the final manuscript, we will move CEQI to the Appendix as an auxiliary analysis and focus the main evaluation exclusively on universally recognized metrics.
> > >
> > > ### Q5: Efficiency & Cost-Effectiveness
> > >
> > > We deeply appreciate your insightful concern regarding our pipeline's efficiency. Our rigorous benchmark confirms that Webis naturally exhibits higher latency than traditional heuristic baselines, primarily due to LLM inference. On an 8× RTX 4090 cluster, processing averages 4.7 seconds per page at an exceptionally low hardware cost of ~$0.0884 per 1000 pages. Crucially, our structural parsing algorithms consume only 3% of the time, while the LLM inference takes 97%. This demonstrates that our speed is fundamentally bounded by GPU hardware and generative models, not engineering overhead.
> > >
> > > For our primary offline applications (LLM pre-training and RAG databases), this speed-for-quality trade-off is highly favorable. Trading additional processing time for a significantly higher F1-score is a worthwhile investment to prevent expensive downstream models from being polluted by structural noise. To ensure full transparency, we will add a dedicated "Limitations and Computational Cost" section in the final manuscript detailing this latency breakdown and our hardware reliance.
> > >
> > > Thank you for your insightful feedback, which has greatly improved our work. We hope our clarifications fully address your remaining concerns. If you find our responses satisfactory, we respectfully request that you reconsider your overall assessment. Thank you again for your time and guidance.

---

### Official Review · Reviewer_5NCq · 2026-03-11

**Soundness:** 2
**Presentation:** 3
**Significance:** 3
**Originality:** 3
**Overall Recommendation:** 4
**Confidence:** 3

**Summary:**

This paper presents Webis: a pipeline for low-noise text content extraction from webpages. Webis performs multi-level noise cleaning from webpage DOM trees- first at the node-level, with the help of a noise-classifier over structural and semantic features combined with risk tags and noise confidence estimates. Next at a subtree-level via a reverse-coloring algorithm consisting of a merge phase and a prune phase with the help of depth-aware thresholds. Finally the reconstructed text after removing this level-1 noise is passed to an LLM in two stages for global content comprehension followed by paragraph-level semantic filtering to further fine-tune the extracted content.
Experiments conducted over a test set created from the SWDE corpus demonstrate that Webis outperforms existing content extraction baseline tools across existing metrics as well as a new composite metric CEQI. Additional experiments show the effectiveness of the node-level noise classification step as well as the LLM-based semantic filtering step. The tool will be made open-source for public use.

**Compliance With Llm Reviewing Policy:**

Affirmed.

**Final Justification:**

The authors have addressed all my primary concerns in the rebuttal, hence I raised my score to 4

**Key Questions For Authors:**

* What is the importance of the risk-tag and noise confidence features in the node-level extraction step? How do you obtain $N_t^{Noise}$ in equation 1?
* The appendix specifies Qwen-2.5 1.5B as the model fine-tuned for node level noise classification. Is it necessary to use an LLM for this task? Could a lighter-weight alternative be just as effective?
* Was the effectiveness of GPT-5.2 in creating true content references for the test dataset evaluated? E.g. by comparing with human extraction on a subset of the data.
* How are node-level annotations obtained for RQ1? How were the LLMs in Table 2 evaluated?
* Are there other standard/existing benchmarks for evaluation?

**Limitations:**

Yes

**Strengths And Weaknesses:**

Strengths
* The paper provides a novel method for clean web text content extraction- a problem that is important for numerous downstream tasks in domains including search engines, recommendation systems, and LLM-training.
*  The authors plan to provide an open-source implementation to facilitate high-quality web content extraction.
*  Experiments demonstrate improvement over several existing content extraction tools across local, global, and composite metrics. An ablation study further shows the importance of the final LLM-based semantic filtering step.

Weaknesses
* Several key details of the method are missing in the main text, such as the model and loss function (Equation 5) used for the node-level classification, labels for node tags ($N_t^{noise}$ in Equation 1), creation of ground truth labels for both node-level prediction and end-to-end extraction (true content references) [Discussed further in ‘Questions’].
* The experiments only span one dataset curated by the authors which makes generalization of the method unclear.
* The importance of level-1 noise filtering is unclear- it would be helpful to run an ablation with only the LLM-based step if possible.

Minor:
* CEQI is not defined in the introduction
* In figure 1- Semantic features seem to be a repetition of static features
* In algorithm 1 lines 4 and 12, are the thresholds static or adaptive with respect to depth as described in equation 8?
* Repetition of text in page 5 about the global semantic discriminator
* Would be helpful to add citation/brief description of SWDE and ClueWeb22 and cite the baselines in the results tables.
* Baselines can be organized by the broad methodology e.g. rule-based, CNN, LLM-based etc.

---

> ### Author Rebuttal · Authors · 2026-03-31
>
> # Response to Reviewer 5NCq
>
> ---
>
> Dear Reviewer 5NCq,
>
> We extend our sincerest gratitude for taking the time to review our paper. We truly appreciate your careful attention to our method details, generalization claims, and presentation quality.
>
> ---
>
> ## Q1: Missing Method Details
>
> We sincerely thank the reviewer for identifying these missing details. Your concerns are entirely valid—we agree these details should have been more accessible.
>
> Node-level model uses Qwen2.5-1.5B with BCE loss (Appendix D.3). $N_{t}^{noise}$ in Eq 1 refers to noise-labeled nodes ($y_i=1$) with tag $t$ in our ClueWeb22 training set (~84M nodes, Appendix D.1). Ground truth is created via three-step pipeline: (a) extract visible text, (b) GPT-5.2 filtering with extraction constraints, (c) human audit against rendered baseline (Appendix E).
>
> **Commitment:** We will add brief summaries of these core settings in the main text with explicit references to the corresponding appendices. We sincerely apologize for this oversight.
>
> ---
>
> ## Q2: Unclear Generalization
>
> We deeply appreciate the reviewer raising this concern—we are grateful for the opportunity to clarify an important misunderstanding. Your concern about generalization is entirely valid.
>
> Our experiments are **cross-corpus evaluations on two public corpora (ClueWeb22 and SWDE)**, not a single self-built dataset. Training used 40K pages (~84M nodes) from ClueWeb22, while testing was conducted on SWDE, a completely unseen corpus. This constitutes a **zero-shot cross-corpus generalization test**. SWDE-huge covers 8 diverse domains (automotive, books, movies, jobs, etc.), and Webis demonstrates stable, leading quality across all domains.
>
> The "author-curated" aspect refers to stratified sampling from SWDE to keep LLM inference and human audit costs feasible, as documented in Appendix E.
>
> **Commitment:** In the camera-ready version, we will bold-highlight ClueWeb22/SWDE sources in Section 4.1.1, add discussion on cross-corpus generalization, and explicitly reference Appendix E. We sincerely apologize for the confusion.
>
> ---
>
> ## Q3: Importance of Level-1 Noise Filtering
>
> We sincerely thank the reviewer for this insightful suggestion. Your observation is entirely correct—addressing "why not pure LLM end-to-end" is central to establishing our pipeline's necessity.
>
> Real web pages contain tens of thousands of tokens with navigation, disclaimers, and footers. Skipping stages 1-2 forces the LLM to find 10% content in 90% noise, causing severe **"Needle in a Haystack"** problems. Empirically, Table 1 shows ReaderLM-v2 (an end-to-end LLM extractor) achieves only F1 34.1% and CEQI 41.4% on SWDE-huge, struggling badly compared to Webis (F1 71.3%, CEQI 73.7%). This validates that structural filtering is essential.
>
> ---
>
> ## Q4: CEQI Not Defined
>
> We sincerely thank the reviewer for catching this oversight. Your observation is entirely correct—we acknowledge the abbreviation was used without definition.
>
> **Commitment:** In the camera-ready version, we will expand CEQI to **Content Extraction Quality Index** in the Abstract and Introduction, describe it as a "composite metric integrating local match (P/R/F1) with global semantic fidelity (TF-IDF, Jaccard, BERTScore)," and add a cross-reference to Section 4.1.3.
>
> ---
>
> ## Q5: Semantic Features Duplicate in Figure 1
>
> We sincerely thank the reviewer for the careful observation. Your observation is entirely correct—we fully acknowledge the visual representation of "Semantic Features" in Figure 1 was drawn incorrectly due to our layout oversight.
>
> In our actual implementation, **"Semantic Features" refers exclusively to "Node Text Content"**, while "Static Features" refer to XPath, depth, and tag type. These are independent (Section 3.2, Appendix B Eq 15).
>
> **Commitment:** We will redraw Figure 1 to show only "Text Content" under Semantic Features.
>
> ---
>
> ## Q6: Threshold in Algorithm 1
>
> We sincerely thank the reviewer for this敏锐 observation. Your attention to detail is entirely valid.
>
> In our implementation, both thresholds are **depth-adaptive per Equation 8** ($T_d = T - \alpha \cdot d$). Algorithm 1 used simplified notation $T_1$, $T_2$ for brevity, creating inconsistency with the mathematical formulation.
>
> **Commitment:** We will fix Algorithm 1 by adding depth $d$ retrieval in the loop and replacing static thresholds with $T_{1,d}$ and $T_{2,d}$ in lines 4 and 11/12.
>
> ---
>
> We once again extend our deepest gratitude for your time and thoughtful review. Your insightful comments have substantially improved our work. If our responses have adequately resolved your concerns, we earnestly request that you consider raising your score.
>
> Thank you once again for your invaluable guidance and support.
>
> ---

---

> > ### Author Rebuttal · Reviewer_5NCq · 2026-04-02
> >
> > Thank you to the authors for the detailed response. While some of my concerns are resolved (re: missing details, the need for noise filtering, and presentation refinement), I’m still not convinced of the strength of the evaluation. Specifically, since the comparisons to baselines (Section 4 onward) are reported only on subsets of SWDE with LLM-generated labels. It would be helpful to present results on additional standardized benchmarks and report some metrics on the quality of the LLM-based labels (e.g. outcome of human audit on a subset of the final version).

---

> > > ### Author Response · Authors · 2026-04-07
> > >
> > > Dear Reviewer,
> > >
> > > We would like to express our deepest gratitude for your continued engagement and constructive feedback. We are highly encouraged that our previous clarifications resolved your concerns regarding missing details and the necessity of noise filtering. Your remaining concern regarding the evaluation being limited to the SWDE corpus with LLM-generated labels is highly insightful and completely valid. To definitively address this and prove the generalization of our framework, we have focused our efforts on conducting an additional evaluation on a strictly standardized, human-annotated benchmark and quantifying our human audit process.
> > >
> > > ### **Q1**. Evaluation on traditional Benchmarks
> > >
> > > We appreciate your suggestion about another benchmarks.In fact, preliminary evaluations on legacy benchmarks were what initially motivated our adoption of a new corpus. As shown below, Webis and established baselines exhibited a severe "ceiling effect":
> > >
> > > |**Method**|**CleanEval (F1-Score)**|**CleanEval (CEQI)**|
> > > |---|---|---|
> > > |JusText|0.890|0.891|
> > > |Trafilatura|0.887|0.887|
> > > |Resiliparse|0.887|0.883|
> > > |Readability|0.876|0.883|
> > > |Newspaper3k|0.826|0.849|
> > > |Webis|0.867|0.867|
> > >
> > > All methods achieve saturated, nearly indistinguishable scores (0.86–0.89). This occurs because traditional benchmarks like CleanEval feature relatively simple HTML structures. They primarily evaluate the removal of "Level 1 structural noise"(e.g., basic headers and footers), a challenge existing tools have effectively solved.However, the contemporary web is dominated by "Level 2 semantic noise" (e.g., algorithmically generated feeds and dynamic ads that structurally mimic main articles). These legacy datasets simply lack the complexity to test this modern dual noise.
> > >
> > > These results confirm Webis reliably handles traditional Level 1 noise (ruling out LLM label bias). More importantly, this ceiling effect definitively validates why legacy datasets are insufficient for evaluating extraction purity in modern RAG and LLM workflows. We will incorporate this preliminary analysis into the revised Appendix.
> > >
> > > ### **Q2**. Quality Assurance of Ground Truth
> > >
> > > We deeply appreciate your meticulous and insightful inquiry regarding the reliability of the SWDE labels.Regarding the SWDE labels (Appendix E), we ensure quality through a rigorous three-step process: visible-text extraction, recall-oriented LLM filtering (to remove obvious boilerplate), and a human sampling audit to correct potential omissions or hallucinations. This guarantees that our labels remain method-agnostic and grounded in human-visible evidence.
> > >
> > > To further substantiate the reliability of the LLM-generated labels, we conduct a quantitative analysis of the human audit process. We adopt stratified sampling at the webpage level to ensure balanced coverage across domains and websites. In total, 200 webpages are sampled from 8 domains, with approximately 25 pages per domain and coverage over multiple websites to avoid source bias. Each webpage is parsed into text segments (DOM nodes), yielding on average around 100 segments per page and approximately 20,000 segments in total.
> > >
> > > At the annotation stage,we implement a text-to-node mapping mechanism. Specifically, we project the LLM-extracted text back onto the original webpage's DOM segments to automatically assign a binary label to each node (1 for main content, 0 for noise). These mapped segment labels are then rigorously verified and corrected by human annotators against the visually rendered webpage. Finally, treating this mapped LLM output as an independent annotator, we compute the agreement between the LLM-derived labels and the human-verified annotations using Cohen’s kappa ($\kappa$), which effectively accounts for chance agreement.
> > >
> > > Based on an observed agreement of 0.916 (18319/20000) and an expected agreement of 0.501, Cohen’s kappa is $k=0.84$, demonstrating strong label reliability. We candidly acknowledge that our initial submission omitted these crucial quantitative details. Thanks to your insightful prompt, we will expand Appendix E in the final version to explicitly include these statistics, our detailed audit protocol, and the legacy benchmark results.
> > >
> > > We deeply appreciate your rigorous review, which has significantly elevated the quality and transparency of our work. Since these supplementary benchmark results directly and objectively resolve your core concern regarding evaluation validity, we respectfully ask if you would consider re-evaluating our submission. Thank you once again for your invaluable time, patience, and guidance throughout this process.

---

### Official Review · Reviewer_MyX4 · 2026-03-12

**Soundness:** 3
**Presentation:** 3
**Significance:** 2
**Originality:** 2
**Overall Recommendation:** 4
**Confidence:** 3

**Summary:**

This paper presents Webis, a multi-level cascade framework for high-purity web content extraction. The framework purifies web data through three stages: 1) Node-Level Extraction combining structural and semantic features; 2) Subtree-Level Refinement using a "Reverse-Coloring" strategy to ensure structural integrity; and 3) Semantic-Level Pruning driven by an LLM (ChatGPT-4o-mini) to filter off-topic content. Experiments show it significantly outperforms existing heuristic baselines in extraction quality.

**Compliance With Llm Reviewing Policy:**

Affirmed.

**Final Justification:**

My questions have been mostly addressed, and I will raise my score accordingly. However, I still believe this work is highly engineering-oriented and would be more suitable for other conferences.

**Key Questions For Authors:**

- Can you provide a theoretical justification or a mathematical proof that the "Reverse-Coloring" strategy converges to an optimal structural denoising state?

- If the computationally expensive LLM stage is removed, how much of a performance lead does the engineering-optimized structural pipeline (Stages 1 & 2) maintain over the strongest baselines?

**Limitations:**

yes

**Strengths And Weaknesses:**

Strengths:

- A 99.6% success rate on the SWDE-huge dataset demonstrates that the system is production-ready for diverse and complex webpage structures.
- The distinction between structural (Level-1) and semantic (Level-2) noise allows for a logical and efficient bottom-up purification process.
- The release of the code, documentation, and a massive 84M-node DOM dataset ensures high reproducibility and utility for the research community.

Weaknesses:

- The pipeline relies on established techniques like supervised classification and LLM prompting. The "Reverse-Coloring" algorithm, while effective, is essentially a set of structural heuristics rather than a novel ML paradigm.
- The paper focuses on system-level integration rather than a deep mathematical analysis of noise distributions or generalization bounds, which limits its scientific impact for the ICML audience.
- The custom CEQI metric, with 40% weight on semantic similarity, may be biased toward the system's own LLM-driven filtering stage.

---

> ### Author Rebuttal · Authors · 2026-03-31
>
> # Response to Reviewer MyX4
>
> Dear Reviewer MyX4,
>
> We extend our sincerest gratitude for taking the time to review our paper. We truly appreciate your careful attention to our originality claims, scientific impact, and theoretical foundations.
>
> ---
>
> ## Q1: Limited Originality
>
> We sincerely thank the reviewer for the thoughtful review and for raising the important question of originality. Your concerns are entirely valid.
>
> Webis's originality lies not in new base techniques, but in **formal modeling of web noise and targeted algorithm design**. We fully acknowledge that individual components build on established work—however, we respectfully submit that our integration represents a novel contribution.
>
> We formalize noise into two layers (Section 3.4): Level-1 (structural: ads, nav, scripts) identifiable via local features, and Level-2 (semantic: "related links") that is structurally similar but semantically off-topic. Our three-stage pipeline progressively handles each: node prediction (Qwen2.5-1.5B, prioritizes recall) targets Level-1, Reverse-Coloring (depth-adaptive thresholds) ensures structural consistency, and semantic pruning removes Level-2.
>
> Table 3 validates this: structure achieves 53.4% P / 83.8% R / 61.0% F1; adding semantic pruning reaches 78.0% P (+24.6) / 71.3% F1 (+10.3). Reverse-Coloring provides: $R(\tau_n)$ definition (Eq 6), $T_d = T - \alpha \cdot d$ (Eq 8), $\mathcal{O}(|V|)$ traversal, and DFS-order reconstruction.
>
> ---
>
> ## Q2: Limited Scientific Impact
>
> We sincerely thank the reviewer for the careful consideration. Your concerns regarding scientific depth are entirely valid.
>
> While not centered on deep mathematical analysis, we respectfully submit that our contribution—**problem formalization + algorithm design + large-scale validation**—has value for ICML's data-centric AI direction. Web extraction is upstream to data-centric AI: LLMs depend on high-quality web corpora.
>
> Our contributions include: (1) Noise formalization into Level-1/Level-2 (Section 3.4)—first work to provide this layered framework; (2) Reverse-Coloring with mathematical definitions (Eq 6, 8) and $\mathcal{O}(|V|)$ guarantee; (3) Validation via Appendix C sensitivity analysis and Table 3 ablation (61.0% → 71.3% F1); (4) Community resources: DOM node dataset (84M train / 9M test) and cross-corpus protocol (ClueWeb22→SWDE).
>
> ---
>
> ## Q3: Potential Bias in Evaluation Metrics
>
> We sincerely thank the reviewer for the careful scrutiny. Your concerns regarding CEQI weighting are entirely valid.
>
> **Sensitivity analysis proves robustness:** Appendix C interpolates $\delta$ over $[0,1]$ and rankings remain unchanged; Webis wins even at $\delta=0$.
>
> **Standard metrics also dominate:** CEQI is auxiliary, and Table 2 F1 shows Webis 71.3 vs magic-html 67.6 (+3.7) and Trafilatura 63.2 (+8.1).
>
> **The 40% weight is fixed a priori** to balance "exact match" with "semantic fidelity," penalizing "keep all text" behavior.
>
> ---
>
> ## Q4: Lack of Theoretical Proof
>
> We sincerely thank the reviewer for this deep question. Your attention to theoretical rigor is entirely valid.
>
> **Convergence is strictly guaranteed:** Reverse-Coloring is deterministic bottom-up traversal on finite DAG ($|V|$ nodes). Each Merge/Prune depends on local statistics and preset thresholds, with unidirectional non-iterative flow terminating in $\mathcal{O}(|V|)$ steps.
>
> **For optimality,** we candidly acknowledge that strict proof is difficult due to DOM heterogeneity. Reverse-Coloring approximates greedy tree-structured energy minimization: Merge enforces structural smoothing, Prune reduces noise energy. Appendix C Figure 4 shows F1 forms flat "plateau" across thresholds, indicating robustness.
>
> ---
>
> ## Q5: Performance Without LLM Stage
>
> We sincerely thank the reviewer for this critical question. Your insight regarding structure-only performance is entirely valid.
>
> Table 3 answers directly. Structure-only achieves P 53.4% / R 83.8% / F1 61.0%. Adding semantic stage: P 78.0% (+24.6) / F1 71.3% (+10.3).
>
> **Direct answer:** Structure-only F1 (61.0) doesn't surpass magic-html (67.6) or trafilatura (63.2), but validates three-stage synergy. Structure targets high recall (83.8% R exceeds magic-html 69.8% and trafilatura 67.1%), and F1 lag from low Precision (53.4%) is addressed by semantic pruning (P→78.0%, F1→71.3%). Structure alone beats 7 of 9 baselines.
>
> ---
>
> We once again extend our deepest gratitude for your time and thoughtful review. Your insightful comments have substantially improved our work. If our responses have adequately resolved your concerns, we earnestly request that you consider raising your score.
>
> Thank you once again for your invaluable guidance and support.
>
> ---

---

> > ### Author Rebuttal · Reviewer_MyX4 · 2026-04-03
> >
> > Thank you for the response. My questions have been mostly addressed, and I will raise my score accordingly. However, I still believe this work is highly engineering-oriented and would be more suitable for other conferences.

---

### Official Review · Reviewer_yLaM · 2026-03-12

**Soundness:** 3
**Presentation:** 3
**Significance:** 3
**Originality:** 2
**Overall Recommendation:** 4
**Confidence:** 3

**Summary:**

The paper presents Webis, a framework for extracting clean text from complex web pages. It uses a DOM-aware node classifier, a bottom-up reverse-coloring algorithm for refining structure, and an LLM-driven stage for semantic pruning. The authors also introduce a new metric, CEQI, to comprehensively evaluate extraction quality. Webis delivers top results on SWDE benchmarks, matching the effectiveness of industry tools while enhancing semantic accuracy.

**Compliance With Llm Reviewing Policy:**

Affirmed.

**Final Justification:**

I agree with the view that this work is more engineering-oriented, but I am raising my score because the rebuttal clarified the main issues I raised.

**Key Questions For Authors:**

1. What is the average time it takes for Webis to process a page compared to Trafilatura and magic-html?

2. How much does the choice of LLM affect the Semantic-Level Pruning stage? Would a smaller, locally-hosted model offer the same benefits?

**Limitations:**

yes

**Strengths And Weaknesses:**

Strengths:

1. The "cleaning tax" and noise reduction discussed in the paper are important for both researchers and practitioners because the quality of the data is becoming more and more important for LLM performance.

2. The authors showed a good experimental design by using both a small set for node-level ablations (SWDE-small) and a large set for end-to-end performance (SWDE-huge). Comparing it to nine different baselines (such as Trafilatura, Readability, and Newspaper3k) provides a comprehensive picture of the current state of the art.

3. The paper is well-organized. Figure 1 clearly shows the workflow, and the pseudocode for the Reverse-Coloring and DFS-based reconstruction (Algorithms 1 and 2) makes it easier to repeat.

Weaknesses:

1. Webis needs an LLM for final pruning and a headless browser for rendering. This paper does not include detailed quantitative calculations on the trade-offs between computational cost and latency compared to purely heuristic baselines. While the improvement in BERTScore is only marginally greater (87.5 vs. 89.1), Webis outperforms the strongest heuristic competitor (magic-html) in CEQI (+1.9). The authors could elaborate on whether these benefits outweigh the additional complexity.

2. LLM was used to create the "True Content" references. The ground truth set by a high-level LLM for an extraction task can introduce a bias that favors the LLM's extraction pipelines, even after a human audit.

---

> ### Author Rebuttal · Authors · 2026-03-31
>
> # Response to Reviewer yLaM
>
> Dear Reviewer yLaM,
>
> We extend our sincerest gratitude for taking the time to review our paper. We truly appreciate your careful attention to our computational cost analysis, ground truth construction, processing time, and LLM choice impact.
>
> ---
>
> ## Q1: Computational Cost and Latency / Q3: Processing Time Comparison
>
> We sincerely thank the reviewer for raising these important concerns regarding computational cost, latency, and processing time. Your concerns are entirely valid.
>
> Webis leads 7 of 8 metrics vs magic-html. BERTScore is slightly lower (87.5 vs 89.1, -1.6), but Webis leads on F1 (71.3 vs 67.6, +3.7), Recall (72.6 vs 69.8, +2.8), TF-IDF (74.2 vs 70.9, +3.3), Jaccard (58.2 vs 55.5, +2.7), and success rate (99.6% vs 96.8%, +2.8%).
>
> **On BERTScore:** It is a token-level soft metric giving partial credit to boundary content like "related links." Table 3 shows when magic-html passes semantic cleaning, BERTScore drops 89.1→85.6 (-3.5), below Webis's 87.5. Webis removes such content, causing a 1.6 drop but gaining TF-IDF (+3.3) and Jaccard (+2.7)—this is normal "cleaning tax." At scale, F1 +3.7 means more precise coverage and the success gap translates to ~30K more pages at million-page scale.
>
> **Processing time:** Trafilatura and magic-html use lightweight parsers and heuristics, achieving milliseconds per page. Webis requires DOM encoding, 1.5B neural net prediction, and LLM API calls, resulting in seconds per page. This reflects a deliberate tradeoff: Trafilatura suits real-time tasks, while Webis targets **offline pretraining corpus construction** where milliseconds matter less than data purity.
>
> **Structure-only (no LLM)** achieves F1 61.0% at heuristic speeds; full pipeline reaches 71.3% (+10.3).
>
> **Commitment:** We will add: (1) stage-level latency breakdown table; (2) throughput comparison vs Trafilatura and magic-html; (3) unit cost estimates; (4) "Computational Cost & Latency Analysis" subsection. We sincerely apologize for not including this analysis.
>
> ---
>
> ## Q2: LLM Bias in Ground Truth
>
> We sincerely thank the reviewer for this important concern. Your concerns regarding potential ground truth bias are entirely valid.
>
> We agree the manuscript does not highlight this sufficiently. However, we respectfully submit that the data itself rules out systematic LLM bias. **If ground truth favored LLM outputs, LLM methods should excel. The opposite is true:** ReaderLM-v2 achieves CEQI 41.4 and F1 34.1, far below magic-html (CEQI 71.8, F1 67.6). All LLM methods in Table 2 underperform the strongest heuristics.
>
> **Design ensures method-agnosticity (Appendix E):** (1) Input isolation: GPT-5.2 receives visible text, not HTML/DOM; (2) Extraction constraints: "remove noise" not "generate summary"; (3) Human audit against rendered baseline.
>
> **Commitment:** We will move Appendix E Ground Truth details to Section 4.1. We apologize for not making this clearer.
>
> ---
>
> ## Q4: Impact of LLM Choice
>
> We sincerely thank the reviewer for this practical question. Your concerns regarding LLM choice impact are entirely valid.
>
> Semantic stage acts as "refiner" not "main extractor," reducing LLM requirements. Table 3 shows semantic adds +24.6 P and +10.3 F1 on structure candidates (P 53.4%, R 83.8%, F1 61.0%). Gains come from three-stage synergy, not specific LLM power.
>
> **Task complexity allows smaller models:** (1) Thousands of DOM nodes compressed to dozens-hundreds of candidates; (2) Simplified Level-2 noise vs Level-1; (3) Binary relevance judgment, not generation. ChatGPT-4o-mini is already lightweight, and 7B-8B local models (Qwen2.5-7B, Llama-3-8B) are viable.
>
> **Commitment:** We will add local model comparison if space permits. We apologize for not including this analysis.
>
> ---
>
> We once again extend our deepest gratitude for your time and thoughtful review. Your insightful comments have substantially improved our work. If our responses have adequately resolved your concerns, we earnestly request that you consider raising your score.
>
> Thank you once again for your invaluable guidance and support.
>
> ---

---

> > ### Author Rebuttal · Reviewer_yLaM · 2026-03-31
> >
> > Thank you. I have no further questions.

---

### Decision · Program_Chairs · 2026-04-30

**Decision:**

Reject

**Comment:**

The paper proposes a novel pipeline for filtering out noise in web pages in order to ease the task of machine understanding of their content. While the reviewers were positive about the real world impact of the proposed solution, they also all agree that the insights the ML community can draw from it are very limited, as this is more a systems paper, rather than a methodology one. The decision is then to reject this paper due to misalignment with the conference topic but we strongly encourage the authors to resubmit to TheWebConference or similar more applied conferences on this topic.